# Bounded rationality in structured density estimation

**Tianyuan Teng**\*
Center for Life Sciences,
Peking University
tengtianyuan@pku.edu.cn

**Li Kevin Wenliang**\*,✉
Gatsby Computational Neuroscience Unit,
University College London
kevinli@gatsby.ucl.ac.uk

**Hang Zhang** ✉
School of Psychological and Cognitive
Sciences, Peking University
hang.zhang@pku.edu.cn

## Abstract

Learning to accurately represent environmental uncertainty is crucial for adaptive and optimal behaviors in various cognitive tasks. However, it remains unclear how the human brain, constrained by finite cognitive resources, internalise the highly structured environmental uncertainty. In this study, we explore how these learned distributions deviate from the ground truth, resulting in observable inconsistency in a novel structured density estimation task. During each trial, human participants were asked to learn and report the latent probability distribution functions underlying sequentially presented independent observations. As the number of observations increased, the reported predictive density became closer to the ground truth. Nevertheless, we observed an intriguing inconsistency in human structure estimation, specifically a large error in the number of reported clusters. Such inconsistency is invariant to the scale of the distribution and persists across stimulus modalities. We modeled uncertainty learning as approximate Bayesian inference in a nonparametric mixture prior of distributions. Human reports were best explained under resource rationality embodied in a decaying tendency towards model expansion. Our study offers insights into human cognitive processes under uncertainty and lays the groundwork for further exploration of resource-rational representations in the brain under more complex tasks.

## 1 Introduction

We study the remarkable ability of humans to learn probabilistic distributions based on experiences in new environments, a skill essential for good performance in a wide range of downstream perceptual [1–4] and cognitive [5–10] tasks. This ability manifests in everyday experiences, such as when a person learns to communicate with others of varying accents

---

\*Equal contributions. ✉Co-corresponding authors. T.T. is now at Max Planck Institute for Biological Cybernetics, Germany. L.K.W. is now at Google DeepMind. H.Z. is also affiliated with Beijing Key Laboratory of Behavior and Mental Health, PKU-IDG/McGovern Institute for Brain Research, and PKU-Tsinghua Center for Life Sciences at Peking University, and Chinese Institute for Brain Research, Beijing.

37th Conference on Neural Information Processing Systems (NeurIPS 2023).

and personalities, or discovers new social norms in foreign cultures. More impressively, humans can not only summarise key characteristics of new experiences but also describe those patterns in statistical terms, such as "Most people in X country typically talk about the weather, but some younger ones prefer to gossip instead". These observations indicate that humans can develop structured and probabilistic internal beliefs about their surroundings from experiences. To distinguish humans' internal beliefs from the modeling framework to be introduced later, we refer to the former as an *internal construct* of the external environment. We ask: how does the human mind form internal constructs from experiences?

As experiences typically arrive sequentially in real life, building a probabilistic internal construct is an online density estimation problem. An ideal solution must be both consistent in distribution estimation and invariant to the order of the experiences. Consistency requires that the learned internal construct approach the true environmental uncertainty as more data arrives, expanding in complexity when the data evidence a more complicated true distribution. Invariance to experience order means no *a priori* preference for any particular model based on the order in which experiences arrive. For instance, if the experiences are independent samples from the same distribution, the learned distribution should not be affected by the sample order. Computationally, these two desiderata can be achieved by choosing an appropriate **internal construct prior** (ICP). Many instantiations of such ICPs incorporate the Chinese restaurant process (CRP) [11–13]. The CRP can recruit more resources to account for more complicated data distributions and is exchangeable so that the order of data does not affect the internal construct. In cognitive science, the CRP has been used to model category learning [14, 15, 6], classical conditioning [5, 16], among other cognitive functions [17].

However, we argue that achieving estimation consistency and order invariance is intractable given humans' limited cognitive capacity. First, buiding a flexible internal construct requires an "infinite capacity" [18] that is at odds with humans' finite memory. The flexibility of the internal construct is likely bounded under resource constraints. Second, achieving order invariance online imposes significant cognitive demands: it requires memorizing all previous experiences and reallocating them into potentially different clusters for every incoming new experience. Indeed, humans do not seem to learn in an exchangeable fashion [19], which can be captured by bounded rational inference [14, 15]. Resource constraints should thus impact the quality of the acquired internal construct.

To investigate how humans actually acquire and represent a structured internal density model under resource constraints, we first designed a structured density estimation task wherein human participants observed sequentially presented samples drawn from a hidden Gaussian mixture with 1–4 clusters. Participants were asked to report the full structure of their internal construct (learned distribution) at the end of the trial, providing us with a high-dimensional behavioral dataset. We found, among others, that participants reported their internal constructs close to the true distribution, but they consistently reported more clusters when there were in fact fewer or even a single cluster in the true distribution.

Motivated by the resource constraint argument, we propose a flexible ICP with a variable propensity towards expansion and includes the exchangeable CRP as a special case. In addition, we fit all parameters of the ICPs to the behavior dataset by a novel density estimation framework (DEF), which allows us to compare between ICPs and explore other inductive biases of participants in this task by likelihood-based mode optimization and critique. Consistent with our hypotheses, the proposed ICP best captures participants' behaviors when it does not preserve order-invariance and instead expands more economically when the internal construct is already complex. Further, fitted ICP explains the inconsistent number of clusters in the reported constrcts as having a strong prior preference for small cluster widths. In sum, our combined experimental and modeling contributions reveal previously unknown human inductive biases in building internal constructs of new environments, and open up doors for further theoretical explorations of flexible priors in learning rich structure representations.

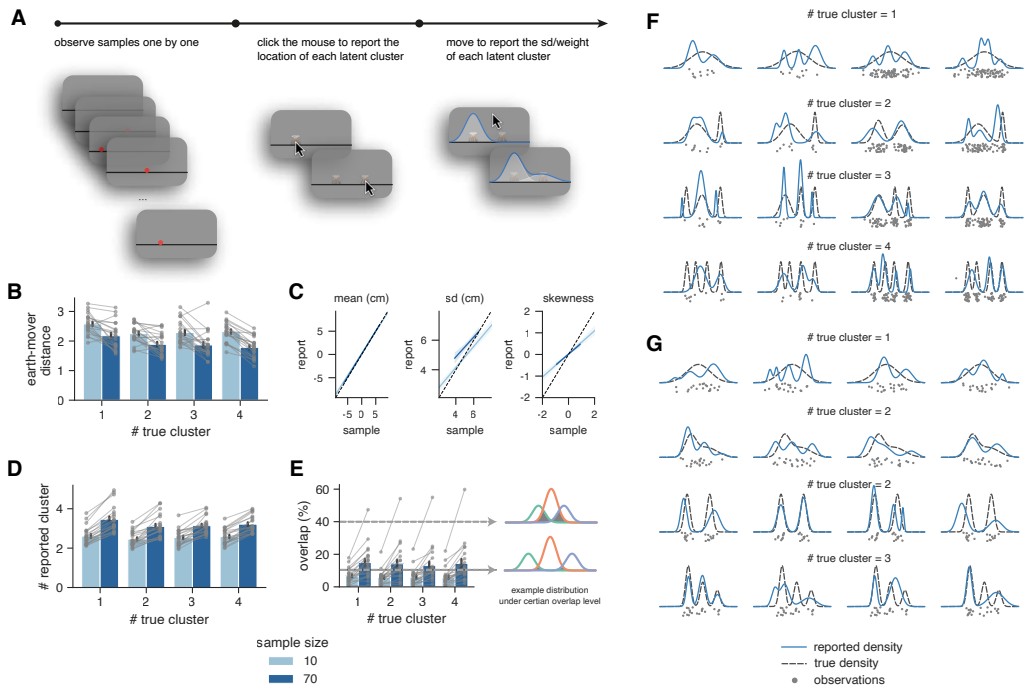

Figure 1: Human behavior in a structured density estimation task. A, On each trial, human participants needed to report the latent Gaussian mixture distribution (i.e., weight, mean, and sd for each cluster) they learned from sequential samples. Panels B–F show the results of Experiment 1. B, The earth-mover distance between the reported and true distributions decreases when participants observe more samples. C, The correlation between the reported and true distributions in the first three moments. See Fig. 6E & F for detailed illustration. D, Mean number of reported clusters. Participants failed to learn the true probabilistic structure, typically reporting 2–3 clusters regardless of the true cluster number. E, Overlap ratio, which is calculated as the percentage of a cluster's area being covered by any other clusters averaged across all clusters in the report. Adjacent clusters did not overlap much. F&G, Participants' reported densities versus the true distributions on example trials of Experiment 1 (F) and Experiment 2 (G). Error bars denote 1 sem over participants. Each gray line in B–D is for one participant.

## 2 Human behavioral experiments of structured density estimation

The human experiments described below were approved by the Institutional Review Board of School of Psychological and Cognitive Sciences at Peking University. All participants received monetary compensation above the local minimum wage.

### 2.1 Experiment 1: learning visuo-spatial distributions

We designed a structured online density estimation task (Fig. 1A) to collect the internal constructs of humans formed from experiences. On each trial, participant saw a sequence of red dots with horizontal positions drawn i.i.d. from a Gaussian mixture with a varying number of clusters across trials (see Appendix A.1 for the cover story and the set of distributions used in each experiment). At the end of the sequence, they were asked to report a Gaussian mixture model that could have generated the red dots, specifying the weight, mean, and variance of each cluster through an interactive interface. As we are interested in participants' innate abilities to build internal constructs, we did not provide feedback.

We first validate the **quality of the reported distribution**. As participants ($N$=21) saw more samples (70 versus 10), the earth-mover distance between the reported and true densities decreased (Fig. 1B, $F(1,20) = 83.21$, $p < 0.001$), which is a desirable trend in density

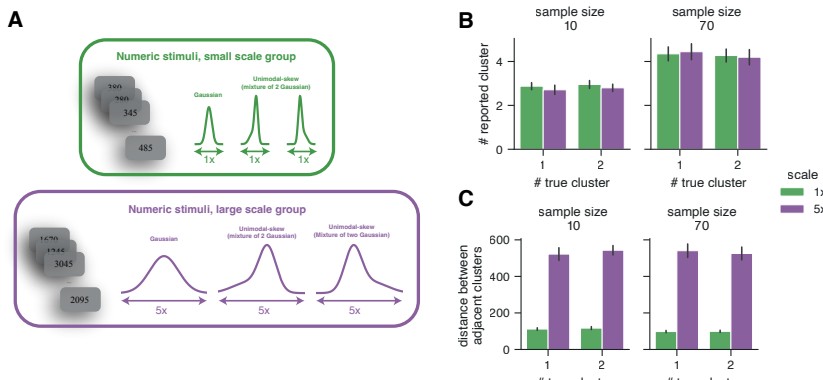

Figure 2: Human learning of numeric distributions. A, Participants were randomly assigned to two groups that differed in the sd of the true distribution (scale 1x vs. 5x). B, The reported cluster number. C, The distance between the adjacent clusters. Error bar denotes 1 sem over participants.

estimation. Figure 1C shows that the reported densities tracked the first three moments reasonably well (all slopes significantly greater than zero, $t$-tests $p < 0.001$), although higher-order moments were harder to estimate.

A closer examination of the reported density on each trial revealed **systematic inconsistencies** in the learned structure. Fig. 1F shows that the reported density had more local maxima even when the true distribution only had a single Gaussian cluster. Participants tended to report around 3 clusters, irrespective of the number of clusters in the true distribution (Fig. 1D). Moreover, with increasing observations, the reported number of clusters increased ($t(20)$=11.15, $p$<0.001) but did not approach the ground truth. Such inconsistencies in the learned number of clusters are striking and counterintuitive, especially when the true generative distribution is as simple as a single Gaussian, in which case more evidence actually pushed participants further away from the truth. Another notable pattern is the small overlap between the adjacent clusters (Figure 1E), which contributes to bumpiness even when the true distribution is as smooth as a Gaussian. Though counterintuitive, this pattern echoes with the finding that humans use near-orthogonal (i.e., non-overlapping) basis functions to represent learned visuo-motor distributions [8].

## 2.2 Experiments 2 and 3: distributions across domains and scales

To show that our findings in Experiment 1 are not specific to the selected distributions, we conducted Experiment 2 ($N$=20) to collect human internal constructs induced by a broader set of 14 representative distributions from previous studies [20–25], with the true number of clusters ranging from 1 to 3. Results in Appendix A.2 show similar findings to Experiment 1: we found reasonably good density estimation quality, and participants preferred to report 2–3 non-overlapping clusters in their internal construct, even when the true distribution is a Gaussian distribution.

Do our findings still hold when we change the domain of the distributions? In Experiment 3 ($N$=36), participants observed samples of numerical values drawn from either Gaussian distributions or unimodal Gaussian mixtures (Fig. 2A). We divided the participants into two groups, one group observed numbers drawn from the original scale ('1X'), and the other observed samples drawn from 5 times the original scale ('5X'). After observing a sequence of numbers, participants, unaware of the true densities, reported the location and weight of each latent cluster. See Appendix A.3 for details.

Consistent with Experiments 1 and 2, we found that participants in both groups of Experiment 3 often reported more complex structures than the ground truth (Fig. 2B). More observations resulted in more clusters ($F(1, 34) = 120.12$, $p < 0.001$). The average reported cluster number was higher (10 samples: 2.8±0.7; 70 samples: 4.3±1.3) than Experiment 1

(10 samples: 2.6±0.4; 70 samples: 3.2±0.5), possibly due to differences in visuo-spatial and numeric memory capacities. Even though the distribution scales differed by a factor as large as five, both groups used a similar number of clusters to represent the numeric distribution ($F(1, 34)=0.05$, $p=0.831$). The distance between adjacent clusters differed roughly fivefold between groups, mirroring the ratio between the true scales (Fig. 2C).

**Summary**   Through comprehensive behavioral experiments, we found that humans were able to build internal constructs of environmental uncertainties based on online experiences, but they could not capture the number of clusters in the true distributions. Learned clusters also tended to be "orthogonal" to each other, creating unsmooth features not present in the true data distribution. It is possible that these imperfections are related to human's limited cognitive resources, as discussed in Section 1, or specific inductive biases in an implicit ICP, such as a small cluster width. Due to the unusual complexity of the density estimation task and the high-dimensional human report, the detailed mechanisms are not directly obvious. We thus resort to building a mathematical model of the reported internal constructs to understand the factors contributing to these inconsistencies.

## 3   The density estimation framework

Our task explicitly queries the internal probabilistic model constructed under sequential experiences. Mapping from these experiences to a distribution function is a challenging and ill-posed problem. Participants need a resource-rational ICP over the space of distributions. In addition, as experimenters, we need an appropriate noise model to define a valid likelihood for maximum-likelihood parameter fitting. To this end, we propose the density estimation framework (DEF) to model the experimental data; it consists of a *rational component* and an *aleatoric component* which we detail below. An overview is shown in Fig. 3.

### 3.1   The rational component

The rational component performs approximate Bayesian inference given data under an ICP, examples of which have been used extensivly in cognitive modelling [15, 16]. A rational ICP for this task factorizes into a prior over the latent cluster assignment of the observed dots, and a prior over the probability density function associated with each cluster. For a sequence of $T$ dot locations $\mathbf{x}_T = [x_1, \ldots, x_T] \in \mathbb{R}^T$, we denote the cluster assignments for $\mathbf{x}_t$ as $\mathbf{z}_t = [z_1, \ldots, z_t]$, and each $z_t \in \mathbb{N}_+$ is the cluster number assigned to $x_t$.

The *economical ICP* we propose here has its prior over $\mathbf{z}_T$ defined recursively as

$$p_{\mathrm{R}}(z_{t+1} = k | \mathbf{z}_t) := \begin{cases} \frac{\tilde{n}_{t,k}}{t+\alpha_t}, & k \leq K_t \\ \frac{\alpha_t}{t+\alpha_t}, & k = K_t + 1 \end{cases}, \quad \alpha_t := \alpha_0 e^{-rK_t}, \quad \tilde{n}_{t,k} := t \frac{n_{t,k}^{\beta}}{\sum_{t=1}^{K_t} n_{t,k}^{\beta}} \quad (1)$$

for $t \in \{1, \ldots T\}$, where $z_1 = 1$, $n_{t,k} := \sum_{\tau=1}^{t} \mathbb{1}[z_\tau = k]$ is the number of clusters assigned to cluster $k$ at time $t$, and $K_t := \sum_k \mathbb{1}[n_{t,k} > 0]$ is the number of non-empty clusters (model size) at time $t$. The expansion rate $\alpha_t$ depends on the model size; for $r < 0$, new clusters are less likely to be added, which implements a form of conservative expansion. The distortion rate $\beta \geq 0$ controls whether the different cluster sizes $n_{t,k}$ are evened-out ($\beta < 1$) or exaggerated ($\beta > 1$), a form of divisive normalization. As a special case, this prior recovers the conventional CRP when $r = 0$ and $\beta = 1$, which is exchangeable and has been a popular choice for modeling flexible online learning. In contrast, the prior (1) is in general not exchangeable; see Appendix B.1.1 for examples. Also, the expected number of clusters is upper bounded by $\sum_t \alpha_0 e^{-rt} < \frac{\alpha_0}{1-e^{-r}}$ for $r > 0$, unlike the conventional CRP where this expectation grows as $\log(t)$. We provide justifications for the non-exchangeability of this prior at the end of this section.

The economical ICP assumes that each cluster is a Gaussian distribution with mean $m$ and variance $v$ as cluster properties. A convenient prior for Gaussians is the normal-inverse-$\chi^2$:

$$p_{\mathrm{R}}(m, v) = \mathcal{N}(m; \mu_0, v/\lambda_0) \mathrm{Inv}\chi^2(v; a_0, \sigma_0), \quad (2)$$

where the hyperparameters with subscript 0 follow standard definitions [15]. Given $\mathbf{x}_T$, Bayesian inference under this ICP defined by (1) and (2) yields a posterior over Gaussian mixture parameters.

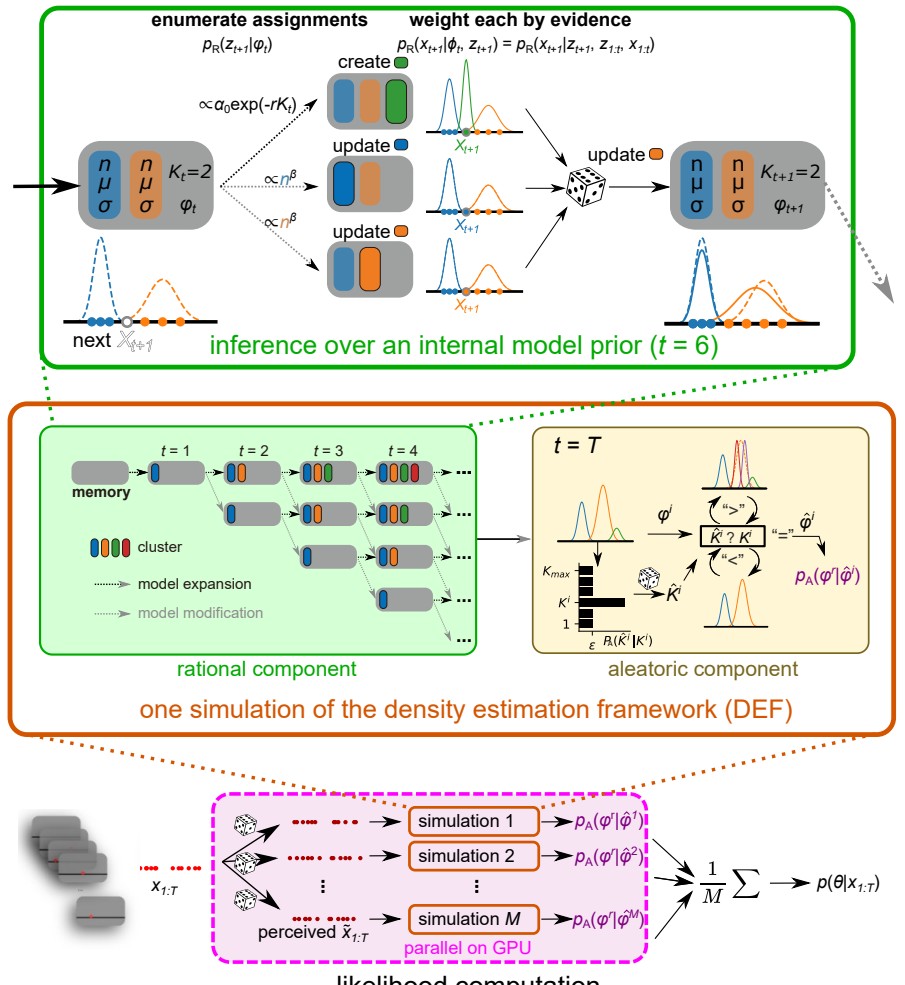

Figure 3: Schematic of the density estimation framework (DEF) and likelihood evaluation. We have omitted the subscripts $t$ and $k$ for clarity. The DEF comprises a bounded rational component (green box) and an aleatoric component (gold box). The former infers the cluster properties $\varphi^i$ based on noisy perceived observations $\tilde{\mathbf{x}}_T$ under an internal construct prior (ICP), and the latter provides a well-defined likelihood function while maintaining the dependencies between the cluster properties. Under the ICP defined in (1) and the particle-based sampling algorithm in (3) and (4), the current density model either updates an existing cluster or adds an additional cluster depending on the evidence of the incoming observation. The evolution of the model corresponds to a single path sampled in the tree structure shown in the left panel of the orange box. In the aleatoric component, structured noise is added to ensure a valid likelihood function for each simulation; see Appendix B. Each prediction yields a conditional likelihood of the reported $\varphi^r$ in each trial, and averaging over a large number of simulations yields the marginal likelihood. These simulations can proceed in parallel (pink box), which can be accelerated by running on GPUs.

Following previous modeling work[15, 5], we assume humans use a single particle to approximately perform the Bayesian updating. Specifically, the observations $\mathbf{x}_t$ up to time $t$ and a corresponding single-particle sequence $\mathbf{z}_t^i$ (an "approximate sample" from the posterior $p_R(\mathbf{z}_t|\mathbf{x}_t)$) define a structured density model: a mixture of $K_t^i = \sum_{\tau=1}^{t} \mathbb{1}[z_\tau^i > 0]$ clusters, where each cluster is weighted in proportion to $n_{t,k}^i = \sum_{\tau=1}^{t} \mathbb{1}[z_\tau^i = k]$, and has posterior

over density given by the usual conjugate prior update formulae:

$$p_{\mathrm{R}}(m_k, v_k | \mathbf{x}_t, \mathbf{z}_t^i) = \mathcal{N}(m_k; \mu_{t,k}^i, v_k/\lambda_{t,k}^i)\mathrm{Inv}\chi^2(v_k; a_{t,k}^i, \sigma_{t,k}^i), \tag{3}$$

$$a_{t,k}^i = a_0 + n_{t,k}^i, \quad \lambda_{t,k}^i = \lambda_0 + n_{t,k}^i, \quad \mu_{t,k}^i = (\lambda_0\mu_0 + n_{t,k}^i\bar{\mu}_{t,k}^i)/\lambda_{t,k}^i,$$

$$\sigma_{t,k}^i = \frac{1}{a_{t,k}^i}\left[a_0\sigma_0 + n_{t,k}^i\bar{\sigma}_{t,k}^i + \frac{\lambda_0 n_{t,k}^i}{\lambda_0 + n_{t,k}^i}(\mu_0 - \bar{\mu}_{t,k}^i)^2\right],$$

where $\bar{\mu}_{t,k}^i$ and $\bar{\sigma}_{t,k}^i$ are, respectively, the empirical mean and the (unadjusted) empirical variance of the observations assigned to cluster $k$ at time $t$, according to the particle $\mathbf{z}_t^i$. Equation (3) shows that, given $\mathbf{z}_t^i$, the sufficient statistics that determine the posterior of the $k$'th cluster are $\boldsymbol{\varphi}_t^i := [n_{t,k}^i, \bar{\mu}_{t,k}^i, \bar{\sigma}_{t,k}^i]_{k=1}^{K_t^i}$, in the sense that the conditioning variables are all functions of $\boldsymbol{\varphi}_t^i$. So, we can equivalently write $p_{\mathrm{R}}(m_k, v_k | \mathbf{x}_t, \mathbf{z}_t^i) = p_{\mathrm{R}}(m_k, v_k | \boldsymbol{\varphi}_t^i)$ in (3). We also regard $\boldsymbol{\varphi}^i$ (not the random variables $m$ and $v$ in (3)) as the inferred cluster properties. When the next observation $x_{t+1}$ arrives, it is assigned to a cluster according to the one-step posterior

$$p_{\mathrm{R}}(z_{t+1}^i | \mathbf{z}_t^i, \mathbf{x}_t, x_{t+1}) \propto p_{\mathrm{R}}(z_{t+1}^i | \mathbf{z}_t^i)p_{\mathrm{R}}(x_{t+1} | \mathbf{z}_t, z_{t+1}, \mathbf{x}_t) = p_{\mathrm{R}}(z_{t+1}^i | \mathbf{z}_t^i)p_{\mathrm{R}}(x_{t+1} | \phi_t^i, z_{t+1}) \tag{4}$$

for $z_{t+1}^i \in [1, \ldots, K_t^i + 1]$, where $p_{\mathrm{R}}(x_{t+1} | \phi_t^i, z_{t+1} = k)$ is the marginal likelihood of $x_{t+1}$ if assigned to cluster $k$, given by evaluating at $x_{t+1}$ the Student's $t$-density with degrees of freedom $a_i$, location $\mu_{t,k}^i$ and scale $(\sigma_{t,k}^i(1 + 1/\lambda_{i,k}))^{1/2}$. If $z_{t+1} \leq K_t^i$, then the $z_{t+1}$'th component is updated by $x_{t+1}$; if $z_{t+1} = K_t^i + 1$, then a new cluster is created, and the internal construct expands. Figure 3 (green box) depicts these two possibilities step at $t = 6$.

If the cluster prior (1) of the rational component implements the CRP, then by exchangeability, the true posterior over the model density is order-invariant with respect to observations. To obtain a particle from an order-invariant posterior, one should revise the cluster assignments of previous $\mathbf{x}_t$ at every time step, but this requires memorizing $\mathbf{x}_t$ which is cognitively implausible. We argue then that keeping the cluster prior (1) exchangeable is unnecessary—it forces the prior to take on a cognitively implausible property. Since (1) subsumes the CRP, we can test whether human behaviors collected in our experiments are better fit by an order-invariant prior.

## 3.2 The aleatoric component and maximum-likelihood model fitting

Note that each inferred particle $\boldsymbol{\varphi}^i := \boldsymbol{\varphi}_T^i$ at $t = T$ is a single simulation of the rational component a participant may possess in their mind. Our (the experimenter's) goal is to fit the rational component to the reported cluster properties $\boldsymbol{\varphi}^r := [\mathbf{w}^r, \boldsymbol{\mu}^r, \boldsymbol{\sigma}^r]$ collecting the weights, means, and variances of the reported clusters; the reported number of clusters implied by $\boldsymbol{\varphi}^r$ is denoted by $K^r$. The key challenges are worth emphasizing: a) the dimensionality of the report is variable, as it depends on the number of clusters; b) the cluster properties are unordered sets, rather than a real vector; c) the cluster assignments $\mathbf{z}_T$ are not observed to us as experimenters and need to be marginalized out to obtain the marginal likelihood, which is intractable. We address the first two challenges by introducing an *aleatoric component*, a structured noise model for DEF; and we deal with the last challenge by an efficient implementation of the DEF that utilizes the parallel processing power of graphical processing units (GPUs).

Figure 3 (right panel of orange box) shows illustrates the aleatoric component. It postulates that the participant, having inferred $\varphi^i$, commits to the number of clusters first, during which they "slack" with a small probability, reporting $\hat{K}^i$ clusters according to:

$$p_{\mathrm{A}}(\hat{K}^i | \boldsymbol{\varphi}^i) \propto \begin{cases} 1, & \hat{K} = K^i \text{ ;} \\ \epsilon, & \hat{K}^i \in \{1, \ldots, K_{\max}\} \setminus K^i \text{ .} \end{cases} \tag{5}$$

where $\epsilon > 0$ is a slack parameter, and $K_{\max}$ is the maximum number of reported clusters seen in an experiment across participants. If $\hat{K}^i \neq K^i$, then the participant removes or splits existing clusters recursively to obtain a set of slacked cluster properties $\hat{\boldsymbol{\varphi}}^i := [\hat{\mathbf{n}}^i, \hat{\boldsymbol{\mu}}^i, \hat{\boldsymbol{\sigma}}^i] =$

$f(\boldsymbol{\varphi}^i, \hat{K}^i)$ until there are $\hat{K}^i$ clusters; Appendix B.2 presents the details of how $f$ modifies $\boldsymbol{\varphi}^i$. Finally, the likelihood of the reported $\boldsymbol{\varphi}^r$ given slacked cluster properties are defined as $P(\boldsymbol{\varphi}^r|\hat{\boldsymbol{\varphi}}^i) := 0$ if $\hat{K}^i \neq K^r$ (infinity penalty for inferring the number of cluster wrong), and

$$p_{\mathrm{A}}(\boldsymbol{\varphi}^r|\hat{\boldsymbol{\varphi}}^i) := \frac{1}{|\mathcal{S}_K|} \sum_{\pi \in \mathcal{S}_K} p_w(\mathbf{w}^r; \pi(\hat{\mathbf{n}}^i)) p_\mu(\boldsymbol{\mu}^r; \pi(\hat{\boldsymbol{\mu}}^i)) p_\sigma(\boldsymbol{\sigma}^r; \pi(\hat{\boldsymbol{\sigma}}^i)) \quad \text{if } \hat{K}^i = K^r, \quad (6)$$

where $\mathcal{S}_K$ is the set of all permutations of $K$ elements; $p_w(\cdot; \mathbf{n})$ is a Dirichlet distribution with concentration defined by $\mathbf{n}$; $p_\mu(\cdot; \boldsymbol{\mu})$ is an isotropic normal with mean $\boldsymbol{\mu}$ and a variance parameter; and $p_\sigma(\cdot; \boldsymbol{\sigma})$ is an isotropic log-normal with mean $\boldsymbol{\sigma}$ and a variance parameter. See Appendix B.2 for precise definitions of these distributions and parameters involved.

Overall, the aleatoric component the takes inferred cluster properties $\boldsymbol{\varphi}^i$ from the rational component and assigns non-zero probabilities to all possible reported distributions through the slack mechanism defined by (5) and the cluster modification $f$. The permutation-invariant likelihood (6) regards the weight, mean, and variance vectors as independent conditional on $\hat{\boldsymbol{\varphi}}^i$, but retains the dependences between the clusters. To approximate the marginal likelihood, we marginalize out $\mathbf{z}^i$ and $\hat{K}^i$ by averaging over a large number of Monte Carlo (MC) simulations. In addition, we allow visual noise by feeding in noisy observations $\tilde{\mathbf{x}}$, where $\tilde{x}_t \sim p_{\mathrm{n}}(\tilde{x}_t|x_t) := \mathcal{N}(\tilde{x}_t|x_t, \sigma_v)$, which is also marginalized out during MC.

Altogether, the DEF estimates the likelihood of the reported $\boldsymbol{\varphi}^r$ given $\mathbf{x}_T$ as $p(\boldsymbol{\varphi}^r|\mathbf{x}_T) \approx$

$$\frac{1}{M} \sum_{i=1}^{M} p_{\mathrm{A}}(\boldsymbol{\varphi}^r|\hat{\boldsymbol{\varphi}}^i = f(\boldsymbol{\varphi}^i, \hat{K}^i)), \quad \hat{K}^i \sim p_{\mathrm{A}}(\hat{K}^i|\boldsymbol{\varphi}^i), \quad \boldsymbol{\varphi}^i \sim p_{\mathrm{R}}(\boldsymbol{\varphi}^i|\tilde{\mathbf{x}}_T^i), \quad \tilde{\mathbf{x}}_T^i \sim p_{\mathrm{n}}(\tilde{\mathbf{x}}|\mathbf{x}_T), \quad (7)$$

where $M$ is the number of MC simulations (see Appendix B.4 for important distinction to the number of particles). We implemented this MC estimator with a fully vectorized approach using PyTorch [26], which enables easy parallelization on GPUs (Fig. 3, pink box). Compared to a conventional for-loop implementation, ours yields reliable log-likelihood estimates with orders of magnitude acceleration. This allows us to optimize the DEF by maximum-likelihood using any off-the-shelve optimization method, such as Nelder-Mead [27]. Further, it also allows us to critique different behavioral models (DEFs) by model comparison, using metrics such as Akaike Information Criterion (AIC) [28].

## 4 Experiments: fitting and comparing internal construct priors

We compare the following ICPs by fitting them in the DEFs that share the same class of aleatoric component: a) our proposed **economical** ICP as described in Section 3.1; b) its special case, the **exchangeable** CRP-GMM; and c) a baseline **batch** learning prior that describes a non-sequential cluster assignment; see Appendix B.1.3 for details. We use the Nealder-Mead optimizer to fit the DEF parameters for each participant, restarting multiple times to avoid early convergence issues [29]. We choose a large number $M$ for MC simulations so that the estimator (7) produces small enough variance and bias tolerable for the Nelder-Mead optimizer. After fitting, we compare different DEFs using a much larger $M$. Details of the fitting algorithm are deferred to Appendix B.3 where we also present additional experiments validating our overall optimization scheme.

The quality of the fitted DEFs on the three experiments are shown in Fig. 4. The economical ICP produced significantly lower AICs than the exchangeable CRP-GMM in all three experiments (Exp. 1: median AICs -2627.9 vs. -2583.1, $p < 10^{-4}$; Exp. 2: -1766.7 vs. -1697.9, $p < 10^{-3}$; Exp. 3: -148.2 vs. -141.0, $p < 0.01$; Wilcoxon signed-rank test). We compare four aspects of the predictions in Fig. 4: the error rate in predicting the number of clusters, the negative log-likelihood (NLL) of the weights, and the expected normalized error in predicting mean and log-variance predictions using the slacked predictions given $\hat{K}^i = K^r$. The advantage of the economical ICP is not only in correctly predicting the number of clusters but also in estimating the cluster properties. As expected, the batch ICP had much worse AICs. Further, to test whether simpler models can explain the data better, we performed an ablation study based on the CRP-GMM ICP (Appendix B.5.1). The ablated ICPs neglect various aspects of the probabilistic structures, effectively mimicking different heuristic

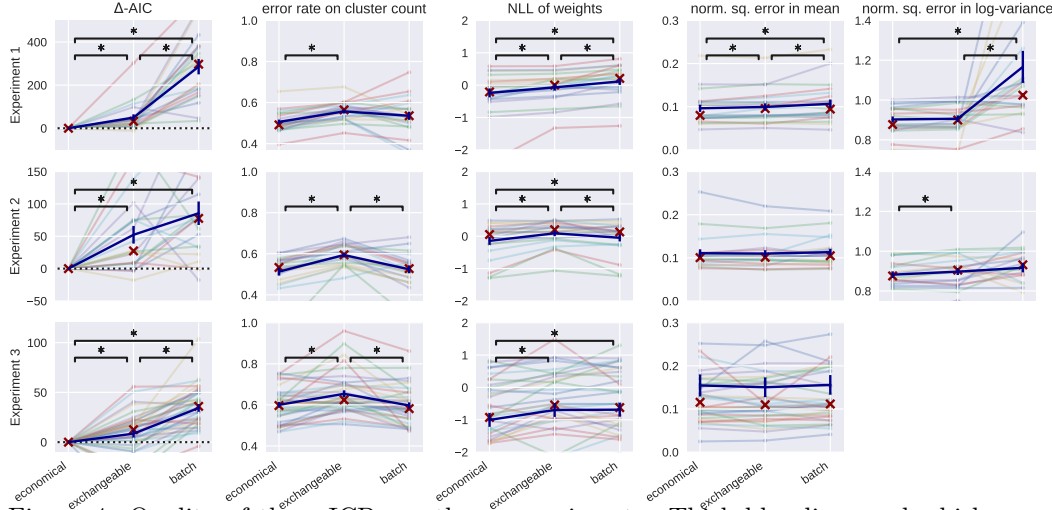

Figure 4: Quality of three ICPs on three experiments. Thick blue lines and whiskers are means $\pm$ 1 sem over all participants (thin lines). Red crosses are medians. Asterisk (*) indicates significance at $p<0.05$ by Wilcoxon signed-rank test.

strategies in forming the internal construct. None of the ablated models outperforms the economical ICP.

Example predictions of the two sequential ICPs are shown in Fig. 5A. The DEF with the economical ICP captures all of the behavioral patterns exhibited by participants, including the inconsistent number of clusters, the low overlap ratio between adjacent clusters, the moments of the reported distribution (Figs. 6 to 8), and the covariance of the reported cluster properties (Fig. 5B-D). We found largely consistent patterns in three fitted parameters across experiments (Fig. 5E–G). First, in line with our resource-rational motivation, the decay rates $r$ in (1) are greater than 0, implying that model expansion is suppressed as $\alpha_t$ decays when the internal construct becomes more complex. Second, participants' prior sd $\sqrt{\sigma_0}$ in (2) was 20%–35% of the true density's sd regardless of the true scale of the observation, with a high confidence $a_0$ (see Appendix B.5.2). This means that each cluster is kept narrow, requiring a new cluster to account for an observation far from existing clusters. It explains the bumpiness in the reported internal constructs, and also suggests that participants adaptively adjusted the cluster scale in accordance with the data scale and modality. Further, the observation that participants reported around 3 clusters is likely the trade-off between the opposite forces of a decaying $\alpha_t$ and a small but confident $\sigma_0$. Third, we found the cluster distortion $\beta$ in (1) to be less than 1, indicating a reduced sensitivity to cluster size when forming the internal construct online.

## 5   Discussion

We studied density estimation by humans, a sophisticated capability fundamental to a variety of cognitive tasks but has so far been elusive in experiments and implicitly assumed in modeling. Through behavioral experiments, we discovered systematic inconsistencies in humans' internal constructs of environments (Section 2) and proposed a density estimation framework for modeling this rich behavioral dataset (Section 3). We explained the inconsistencies as a result of an economically-expanding internal construct prior (ICP) over internal constructs and a strong preference for narrow clusters [30]. The scale-adaptive cluster prior aligns with prior research on adaptive sensory and probability coding [31–33, 30, 34].

The density estimation framework (DEF) is a generic approach for modeling complex behavioral data. Thanks to the generality, one can compare other inference algorithms based on likelihoods and can easily incorporate priors over parameters (hyperpriors). While previous work kept the trainable parameters small and/or used error-based objective functions [15, 35, 4], our DEF can fit more parameters by maximum-likelihood, marginalizing the

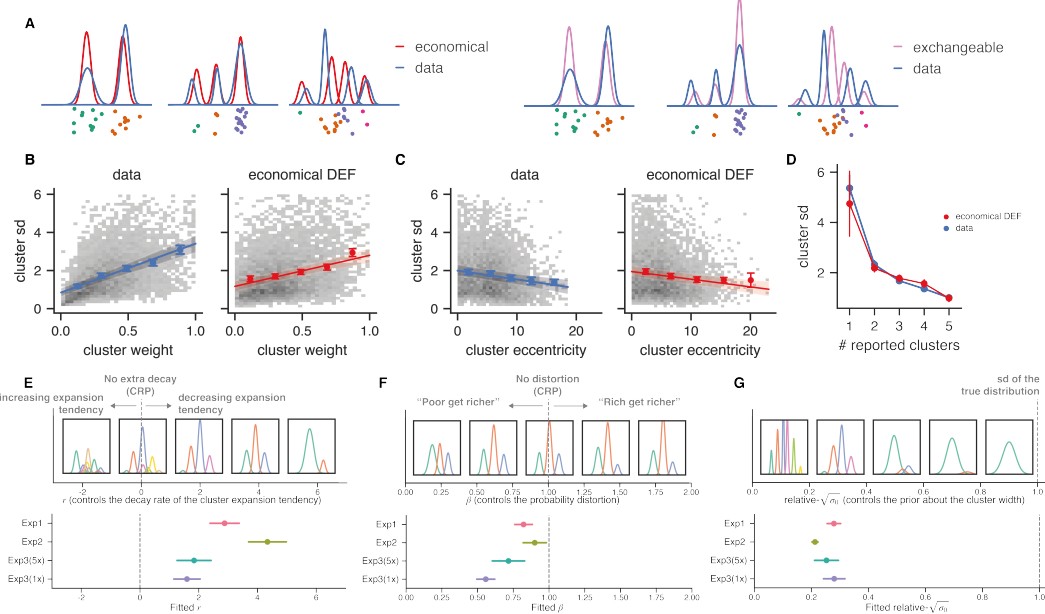

Figure 5: A, Example DEF predictions for Experiment 2 when the predicted number of clusters agree with human reports. B–C, Detailed patterns of the reported cluster properties in Experiment 2, which can be successfully captured by economical DEF. In each panel, the grey heatmap denotes the distribution of responses of individual trials (collapsed across participants). The colored dots and error bars denote average response and standard error across participants in local data bins. The line denotes the regression line. B, the cluster sd increased with the cluster weight. C, The cluster sd decreased with the cluster eccentricity (distance between the cluster center and the global center). D, The cluster sd decreases with the number of reported clusters. E-G, Upper panels show the effects of key parameters in the economical ICP; lower panels show the parameters fit on the data of the three experiments. More visualizations of DEF predictions appear in Figs. 6 to 8 in Appendix A

latent cluster assignments properly, thanks to our efficient implementation and hardware acceleration. With this methodology, we showed that the economical ICP provided a closer match to human behavior than the Chinese restaurant process (CRP) which has been used pervasively in modeling online learning. With a decaying rate of expansion, the growth of an economical ICP is more in line with humans' finite memory capacity.

Our work sends important messages to the theoretical community. Dasgupta and Griffiths [32] showed that the CRP is cognitively plausible because it can be derived from a preference for low entropy in the cluster assignment distribution. We found that the fitted $\beta < 1$ actually increases the entropy of the prior cluster weight distribution [20, 23], suggesting that the cluster count might be the main consumption of cognitive resources. Gershman et al. [16] discussed in great depth on batch versus online learning, whether the model capacity should be finite or infinite, and the hypothesis that retrospective cluster reassignment may be possible with particle representations. We provide key experimental and modeling evidence that humans may employ an online and yet finite (in expectation) model. However, with only a few number of particles, the beliefs of the other possible assignments are not well retained, so retrospective corrections may not be easily produced from these models.

In sum, the economical DEF reproduces many human patterns, not only in the loss function optimized for (quality of capturing human structure learning), but also in the moments of the whole distribution and the covariance structures of cluster properties. We note the large room for improvement in predicting the cluster variance (the right column in Fig. 4), which hints at the complexity of the task and the challenge in modeling. Our work also provides a comprehensive likelihood-based model comparisons on human density estimation, and paves way for future studies using high-dimensional reports in complex behavioral tasks.

## Acknowledgments

This work was partly supported by the National Natural Science Foundation of China (32171095), National Science and Technology Innovation 2030 Major Program (2022ZD0204803), and funding from Peking-Tsinghua Center for Life Sciences to H.Z. L.K.W. was supported by Gatsby Charitable Foundation. We thank Peter Dayan and Jian Li for helpful discussions, and five anonymous reviewers for their valuable feedback.

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

# Bounded rationality in structured density estimation: Supplementary material

## A   Experimental details

### A.1   Experiment 1

#### A.1.1   Participants

Experiment 1 recruited 21 participants (11 females, aged 18–25). All participants had provided informed consent before the experiment.

#### A.1.2   Cover story

Participants were told that they were apprentice magicians in a magical world. In this world, dangerous magic lava rocks were emitted from an unknown number of invisible volcano(es). On each trial, they observed past landing locations of lava rocks in a specific area (on the screen), and their job was to predict the probability density of future landing locations. More specifically, they were asked to draw a probability density by reporting, using click-and-drag mouse gestures, three key properties of the volcano(es), corresponding to the mean, the weight, and the standard deviation of a Gaussian component. They were told that their bonus payment depended on the accuracy of the reported predictive density.

#### A.1.3   Procedure and Design

On each trial, the landing positions of lava rocks were visualized as red dots and sequentially presented on a black line. The number of presented rocks (i.e., sample size) had two levels: 10 or 70. Each dot was presented for 166 ms, followed by a 166 ms empty screen. The landing positions were i.i.d. samples drawn from an unseen mixture distribution.

After observing all dots, participants needed to follow a two-stage procedure to report both (1) the predictive density of rocks' future landing positions, and (2) the underlying generative model. First, they marked the location of the volcano(es) by clicking on the black line. After marking all volcano(es), they pressed "F" to proceed to the next reporting stage. In the second stage, they chose a volcano by clicking the volcano icon and then moved the mouse to report the relative density of the landing position of future lava rocks emitted from this volcano. During the report, the overall density of the rock landing position was computed and presented in real-ime.

The true generative distribution set was composed of 24 distributions (Fig. 6A). The distributions were created by following a merge-from-four procedure. All distributions' overall standard deviations were about 5.3 cm. On each trial, a uniform jitter ([-5.3 cm, +5.3 cm]) was added to the mean of the true generative distribution.

Participants completed 2 (sample size levels) × 4 (true cluster number levels) × 6 (distribution subtype) × 3 (number of repeats) = 144 trials in total. The experiment length was about 105 minutes.

#### A.1.4   Generation of true distribution set

The set of true distributions was constructed following the steps below. First, we created 6 four-cluster Gaussian mixture distributions (bottom row in Fig. 6A) in which each component had the same SD of 0.72cm of visual angle and equal weight. The three center-to-center distances between their adjacent Gaussian components, denoted $[d_1, d_2, d_3]$ (from left to right, measured in cm), were chosen from the set of all permutations of $\{3.6, 4.65, 5.7\}$cm.

Second, each of the 6 four-cluster Gaussian mixtures went through "merging steps", inspired by the proposal step in the dynamic clustering algorithms (e.g. Reverse-jump MCMC) in statistics [36]. In each merging step, we chose two adjacent Gaussian components to merge into one, with the post-merger new component having the same zeroth, first and second

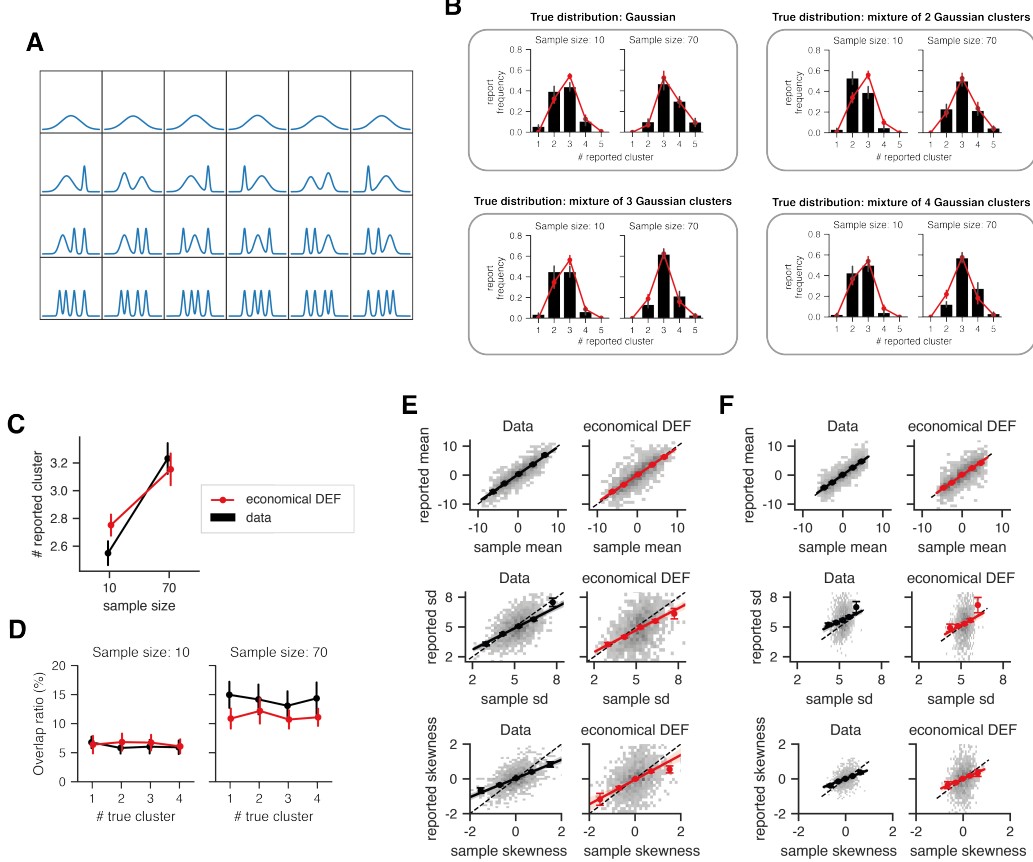

Figure 6: Detailed experimental design and results in Experiment 1. A, The distribution set. B–F shows the prediction of the fitted economical DEFs compared to participants' reports under different conditions. B, The relative frequency of the reported cluster number. C, The average reported cluster number. D, The overlap ratio between the reported clusters. E, The reported moments versus the sample moments in the 10 sample-size condition, with data (left sub-panels) contrasted with model prediction (right sub-panels). Three rows are for mean, sd, and skewness. In each panel, the grey heatmap denotes the distribution of responses of individual trials (collapsed across subjects). The 5 dots and error bars denote average response and standard error across subjects in 5 local data bins. The line denotes the regression line, with shading representing 95% confidence interval. F, The reported moments versus the sample moments in the 70 sample-size condition

moments as the combination of the two pre-merger components. The to-be-merged adjacent components were chosen in such a way that the post-merger Gaussian mixtures minimize KL-divergence $\mathrm{KL}(p_{\mathrm{pre}}||p_{\mathrm{post}})$. By applying the merging step iteratively, the original four-cluster Gaussian mixtures were transformed into three-cluster, two-cluster, and finally one-cluster mixtures.

### A.1.5 Results

See Fig. 6 for the prediction of the fitted economical DEF.

## A.2 Experiment 2

Experiment 2 recruited 21 participants (13 females, aged 18–25). All participants had provided informed consent before the experiment. One participant was excluded due to

Table 1: The distribution set in Experiment 2

| Distribution | Paper | Skewed or Symmetric | Number of clusters | Number of modes |
|:---:|:---:|:---:|:---:|:---:|
| 1 | [24] | Symmetric | 1 | Unimodal |
| 2 | [24] | Symmetric | -* | Unimodal |
| 3 | [25] | Symmetric | 2 | Unimodal |
| 4 | [21] | Skewed | 2 | Unimodal |
| 5 | [21] | Skewed | 2 | Unimodal |
| 6 | [21] | Skewed | 2 | Unimodal |
| 7 | [25] | Skewed | 2 | Unimodal |
| 8 | [23] | Skewed | 2 | Bimodal |
| 9 | [25] | Symmetric | 2 | Bimodal |
| 10 | [22] | Symmetric | 2 | Bimodal |
| 11 | [22] | Symmetric | 3 | Trimodal |
| 12 | [20] | Skewed | 3 | Trimodal |
| 13 | [20] | Skewed | 3 | Trimodal |
| 14 | [20] | Skewed | 3 | Trimodal |

* This is a uniform distribution with smoothed edges.

their task performance being an outlier (measured by the earth-mover distance between the reported predictive density and the true density, exceeding 3 standard deviations, z-score = -3.3).

Experiment 1 uses 14 representative Gaussian mixture distributions selected from previous studies (Table 1). The distribution set can be divided into four subtypes: unimodal-symmetric, unimodal-skewed, bimodal, and trimodal. Note that the number of modes is not necessarily equal to the number of latent Gaussian clusters. For example, a unimodal skewed distribution could be a mixture of two latent Gaussian clusters.

In each trial, participants observed 20 sequential samples drawn from one of the 14 distributions. The standard deviations of all distributions were rescaled to 4.8 cm. Each dot was presented for 166 ms, followed by a 166 ms empty screen interval. On each trial, a common uniform jitter ([-3.7 cm, +3.7 cm]) was added to the means of the clusters. To balance the skewness of the distribution in the experiment, we horizontally flipped the distribution in half of the trials. To explore participant's consistency in structure learning, we presented the same sample sequence for two times in seperate trials. Participants completed a total of 14 (distribution types) × 2 (flip or not) × 2 (number of random sequences) × 2 (number of repeats) = 112 trials. The experiment lasted about 90 minutes.

### A.2.1 Results

See Fig. 7B–D&G for the prediction of the economical model.

Experiment 2 contained repeated trials with identical stimuli. Using these trials, we show in Fig. 7E that in repeated trials participants reported a different number of clusters just below 50% of the time. Similarly, our model could predict with an accuracy around 0.5, very close to the participants themselves. This shows that our model can predict human report close to the participants themselves.

To illustrate the robustness of the model fitting procedure, we run model recovery experiments. Given randomly chosen parameters for the full model, we generate 100 sets of synthetic stimuli, reset the parameters to new random values, and then fit the parameters on the synthetic dataset using the procedure described in the main paper. The results show that the recovered parameters are largely consistent with the random initial values (Fig. 7F, the average correlation between the source parameters and the fitted parameters is 0.84).

### A.3 Experiment 3

Experiment 3 recruited 36 participants (21 females, aged 18–26). All participants had provided informed consent before the experiment.

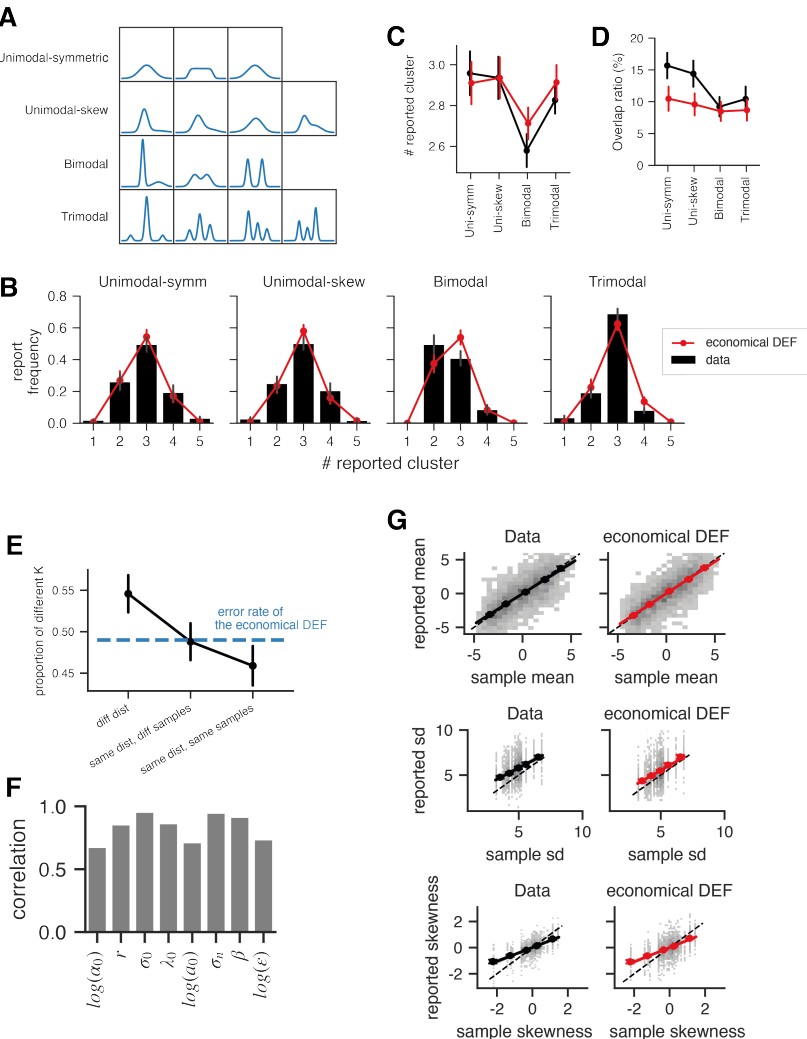

Figure 7: Detailed experimental design and results for Experiment 2. A, The distribution set. B-D&G shows the prediction of the fitted economical DEFs compared to participants' reports under different conditions. B, The relative frequency of the reported cluster number. C, The average reported cluster number. D, The overlap ratio between the reported clusters. E, The proportion that the reported numbers of clusters are different when two trials have (1) different distributions, (2) same distribution but different samples, or (3) same distribution and samples. F, The quality of model recovery, measured by the correlations between the source parameters and the fitted parameters. G, The reported moments versus the sample moments.

In Experiment 3, participants needed to learn the distribution of numeric values and report their belief of the distribution by entering numbers on the keyboard. On each trial, the horizontal coordinates of lava rocks were shown one-by-one on the screen, with each coordinate presenting for 1.5 seconds, followed by a 0.5-second empty screen. After observing all coordinates, participants were required to first report the number of volcanoes. Then, they were required to enter the location and the relative eruption frequency of each volcano.

The true distribution set was composed of 2 unimodal distributions: one is a Gaussian distribution, and the other is a skewed Gaussian mixture with 2 wide clusters (Fig. 2A). Participants completed 3 (distribution type) × 8 (number of repeats) = 24 trials in total.

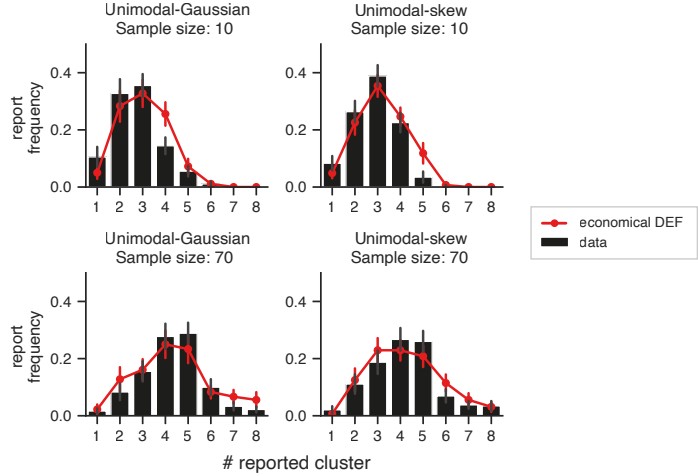

Figure 8: The relative frequency of the reported cluster number in Experiment 3.

### A.3.1 Results

See Fig. 8 for the prediction of the economical model.

## B Density estimation framework details

### B.1 Rational component

The rational component should take the sequential observations $\mathbf{x}_T$ and predict the internal construct properties $\boldsymbol{\varphi}^r$ reported by a participant. From a computational perspective, any simulation-based model could be used for this component, as long as it can produce an internal construct based on $\mathbf{x}_T$. In this work, we adopt the rationale of Bayesian inference given a plausible generative process that could have produced the observations $\mathbf{x}_T$, with approximations mandated by cognitive constraints. Since the behavioral report is derived from a subjective belief of the unobserved distribution in the environment, the generative process in the participant's mind describes a subjective prior (or a construct) over distributions. As such, we call this generative process an "internal construct prior" (ICP).

The ICP in our main model is inspired by the nonparametric Gaussian mixture model. A popular instantiation of the said model is defined through a CRP prior over the cluster assignments, and a conjugate prior over the distribution of each cluster, abbreviated as CRP-GMM. It is appealing for our purpose because the implied generative process of $\mathbf{x}_T$ is sequential, similar to how participants observe the $\mathbf{x}_T$. To make this prior more flexible, we extend the cluster assignment prior by introducing additional parameters in (1) to control the expansion decay rate ($r$) and count distortion rate $\beta$. The effects of these two parameters on the prior distribution of partitions are shown in Fig. 9. While the number of clusters in the ordinary CRP grows, the decaying parameter $\alpha$ in our extension substantially slows down the introduction of new clusters as the model size increases. As of the distortion rate $\beta$, when $\beta < 1$, there is weaker rich-get-richer effect, giving more evenly distributed cluster sizes and higher entropy in the cluster assignment distribution, while a $\beta > 1$ produces more extreme cluster sizes and lower entropy of the cluster assignment distribution.

### B.1.1 Inexchangeability of the the economical ICP

The exchangeability of a cluster assignment prior is often desirable for online data modeling, because it ensures that the posterior of cluster assignment is invariant to the order (permutation) of data. However, in this work, we are interested in a prior that governs how humans construct a density model online. Our discussion on cognitive constraints in

Section 1 suggests that achieving order invariance requires implausible computations, so we propose to relax order invariance for the purpose of modeling human cognition.

Here we show by counterexamples the proposed ICP cluster assignment prior (1) is not exchangeable. First, for the case $\beta = 1$ and $r \neq 0$. consider the partition $\{\{z_1, z_2\}, \{z_3\}\}$. We have

$$\mathbb{P}(z_1 = 1, z_2 = 1, z_3 = 2) = \mathbb{P}(z_1 = 1)\mathbb{P}(z_2 = 1|z_1 = 1)\mathbb{P}(z_3 = 2|z_1 = 1, z_2 = 1)$$

$$= 1\frac{1}{1 + \alpha_0 e^{-r}}\frac{\alpha_0 e^{-r}}{2 + \alpha_0 e^{-r}},$$

which is not the same as the probability of a permuted but equivalent partition,

$$\mathbb{P}(z_3 = 1, z_1 = 2, z_2 = 2) = \mathbb{P}(z_3 = 1)\mathbb{P}(z_1 = 2|z_3 = 1)\mathbb{P}(z_2 = 2|z_3 = 1, z_2 = 2)$$

$$= 1\frac{\alpha_0}{1 + \alpha_0}\frac{1}{2 + \alpha_0 e^{-r}}.$$

Then, for the case $\beta \neq 1$ but $r = 0$, consider the partition $\{\{z_1, z_3, z_5\}, \{z_2, z_4\}\}$. We have

$$\mathbb{P}(z_1 = 1, z_2 = 2, z_3 = 1, z_4 = 1) = 1\frac{\alpha_0}{1 + \alpha_0}\frac{2 \cdot \frac{1^\beta}{1^\beta + 1^\beta}}{2 + \alpha_0}\frac{3 \cdot \frac{2^\beta}{2^\beta + 1^\beta}}{3 + \alpha_0} = \frac{3\alpha_0 2^\beta}{(2^\beta + 1)\prod_{t=1}^3 (t + \alpha_0)},$$

which is not the same as the probability of a permuted but equivalent partition,

$$\mathbb{P}(z_1 = 1, z_3 = 1, z_4 = 1, z_2 = 2) = 1\frac{1}{1 + \alpha_0}\frac{2}{2 + \alpha_0}\frac{\alpha_0}{3 + \alpha_0} = \frac{2\alpha_0}{\prod_{t=1}^3 (t + \alpha_0)}.$$

Therefore, the prior defined by (1) is not exchangeable in general if $\beta \neq 1$ or $r \neq 0$.

### B.1.2 Sufficient statistics

For completeness, we give the explicit expressions of the sufficient statistics in (3) as

$$n_{t,k}^i = \sum_{\tau=1}^t \mathbb{1}[z_\tau^i = k], \quad \bar{\mu}_{t,k}^i = \frac{1}{n_{t,k}^i}\sum_{\tau=1}^t \mathbb{1}[z_t^i = k]x_t, \quad \bar{\sigma}_{t,k}^i = \frac{1}{n_{t,k}^i}\sum_{\tau=1}^t \mathbb{1}[z_t^i = k](x_t - \bar{\mu}_{t,k}^i)^2. \quad (8)$$

We assume that participants maintain these sufficient statistics when building their internal constructs, and also report them at the end of the trial, after normalizing $\mathbf{n}^i$ to obtain the weights. Note that, unconventionally, we denote the *variance* by $\sigma$.

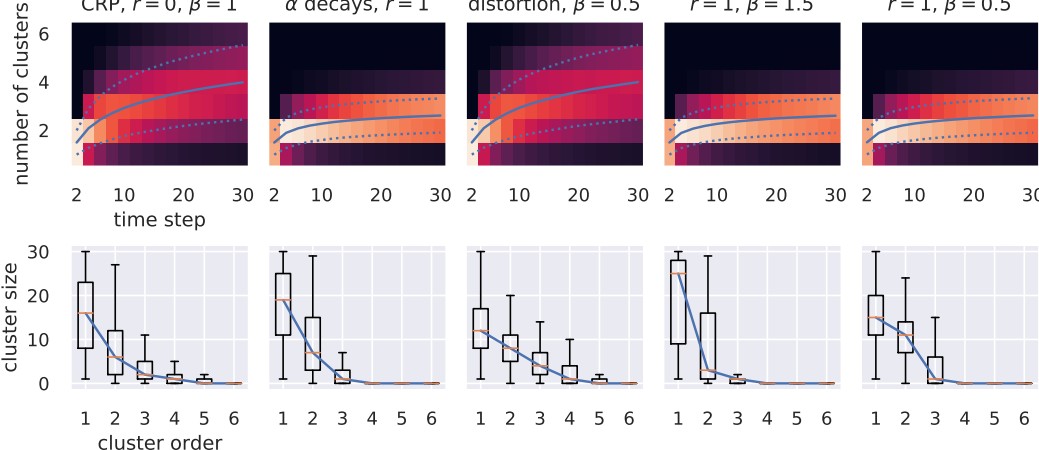

Figure 9: Effects of the decay rate and distortion rate $\beta$ in (1) on the distribution of partitions. Heatmaps on the top row show distributions of the number of clusters as a function of time steps. Brighter color means higher probability. Blue solid line indicates the mean, blue dotted lines indicate 1 standard deviation. The bottom row shows the distribution of cluster sizes at $t = 30$. Yellow bar indicates median.

### B.1.3 Baseline batch ICP

As both the economical ICP in Section 3.1 and the proposed fitting algorithm (to appear in Appendix B.3) are new, it is important to compare the economical ICP and the exchangeable ICP, CRP-GMM, with another baseline ICP, all fit by the same algorithm. If we expect the baseline ICP to provide a worse description of human cognition than CRP-GMM on this task, then the proposed algorithm must be able to produce a less favorable model comparison result for this baseline after fitting them to human data. We choose a batch ICP as the baseline: it generates all observations as i.i.d. samples conditioned on cluster assignments, but, unlike the CRP, the cluster assignment prior is time-invariant and is thus exchangeable; inference over this ICP produces order-invariant posteriors. Because our task has a strong sequential nature, we expect this batch ICP to be a worse descriptor of human cognition for this task.

Specifically, this batch ICP $p_B$ maintains a truncated Poisson prior distribution over $K_{\max}$ submodels of mixture distributions.

$$p_B(K) \propto \begin{cases} \text{Poisson}(K; \bar{K}), & 1 \le k \le K_{\max}; \\ 0, & \text{otherwise.} \end{cases} \tag{9}$$

The $K$'th submodel is a Gaussian mixture of $K$ components. Each submodel has a symmetric Dirichlet prior over the cluster weights, and each component has mean and variance following the conjugate Gauassian-Inverse-$\chi^2$ distribution, as in (2). The truncated Poisson prior is a heuristic choice; we also tried other priors, such as a Categorical distribution supported on $K = 2$ and $K = 3$, or a truncated Poisson with mean dependent on the sample size (10 vs 70). These choices improved the AIC only insignificantly.

**Inference in each submodel** The following description applies to each of the $K_{\max}$ submodels. Given the observed noisy data $\tilde{\mathbf{x}}_T^i$, each submodel produces an inferred cluster property $\varphi_K^i$, which are the sufficient statistics of the cluster property posteriors returned by the variational-EM algorithm (see Chapter 10.2 of [37]). We now drop the superscript $i$ to reduce clutter, bearing in mind that all the variables are samples depending on a noisy observation $\tilde{\mathbf{x}}_T^i$ and a sampled $K^i$. This algorithm alternates between the E-step: updating a Dirichlet posterior over the cluster weights; and the M-step updating a Gaussian-Inverse-$\chi^2$ posterior over the cluster means and variances. It turns out that the posterior over the cluster parameters $\varphi_K$ depends on a set of sufficient statistics similar to (8), except that the cluster assignments are now soft as computed by the responsibilities $r_{t,k}$ for each $x_t$ and $k \in \{1, \ldots, K\}$ during the variational E-step.

After the variational-EM procedure converges, we perform a hard cluster assignment for each observation $x_t$ by choosing the cluster with the highest responsibility,

$$z_t = \arg\max_{k \in \{1, \ldots, K_{\max}\}} r_{t,k}.$$

This ensures that the cluster weights are quantized, as is the case for the particle-based inference methods on the other two ICPs—we do not want to introduce additional effects to the variational model due to having continuous support for cluster weights. Thus, at the last time step $t = T$, the cluster weights are determined by $n_k = \sum_{t=1}^{T} \mathbb{1}[z_t = k]$. The cluster mean $\mu_k$ and variance $\sigma_k$ still depend on the original unquantized responsibilities and per-cluster sufficient statistics similar to (8) but with indicators $\mathbb{1}[z_t = k]$ replaced by responsibilities $r_{t,k}$. As a result, for each submodel with $K$ clusters, the inferred cluster properties are summarized by $\varphi_K := [n_k, \bar{\mu}_k, \bar{\sigma}_k]_{k=1}^{K}$.

**Submodel selection** After obtaining sufficient statistics for all $K$ models, the batch rational component selects the submodel with the largest marginal likelihood given the hard cluster assignments, computed through Student's $t$ marginal densities as in (4), minus a penalty of a weighted model size

$$K_B := \arg\max_{K \in \{1, \ldots, K_{\max}\}} \{p_B(\mathbf{x}_T | \hat{\varphi}_K) - \gamma K\} \tag{10}$$

where $\gamma$ is a trainable parameter. This penalty is consistent with how AIC penalizes model complexity. Then, the submodel with $K_B$ clusters is selected, passing $\varphi_{K_B}$ to the aleatoric component. As such, the batch-based rational component differs from the CRP-GMM in the following ways:

1. In the batch rational component, the cluster assignments are performed by variational inference rather than a particle filter;

2. the batch rational component performs model selection that penalizes large models after inferring multiple submodels, whereas CRP-GMM embeds the preference for smaller models in the cluster assignment prior.

**Likelihood approximation**  These inferred cluster properties are passed to the aleatoric component, giving (slacked) number of cluster $\hat{K}_B$ and the predicted $\hat{\varphi}_B$. This amounts to a single simulation of the batch DEF. To compute the likelihood, the batch DEF still needs to marginalize out the visual noise and the slacked $\hat{K}_B$ in the aleatoric component. To this end, we run a large number of simulations, each with an independent draw of noisy observations $\tilde{\mathbf{x}}_T^i$, giving the predicted cluster properties $\hat{\varphi}_B^i$. The likelihood $p(\varphi^r|\mathbf{x}_T)$ for the batch model is then approximated by (7).

The batch rational model and the aleatoric component combine to give the batch DEF, which is used to benchmark the economical ICP and exchangeable ICP in the corresponding DEFs.

## B.2   Aleatoric component

Here, we explain in more detail the aleatoric component described in Fig. 3. As discussed in Section 3.2, one challenge in modeling this dataset is that the dimensionality of the internal construct varies across trials. This requires that the model be able to produce variable-dimensional predictions. In order to fit the DEFs by maximum-likelihood, the DEF must place nonzero probability to all possible numbers of clusters. We thus defined the distribution $p_A(\hat{K}|K)$ in (5) supported on $\{1, \ldots, K_{\max}\}$, and restrict the maximum number of clusters to $K_{\max}$ to be the largest number of clusters ever reported by participants in an Experiment. One can also define other slack distributions over $K$ with decaying tails to avoid an explicit upper bound.

If there is no slack, then the predicted cluster properties are as inferred. If the participant slacks with probability $\epsilon$ and commits to a prediction $\hat{K}^i$ that does not agree with the inferred $K^i$ from the rational component, they must modify the inferred $\varphi^i$ so that there are $\hat{K}^i$ clusters. We assume that, during modification, the participant should keep the overall distribution roughly intact. We propose the following deterministic procedure, denoted by $f(\varphi^i, \hat{K}^i)$, which recursively increases or decreases the number of clusters in $\varphi^i$ until there are $\hat{K}^i$ left.

**Removing the smallest cluster.**  This happens whenever $\varphi^i$ has more clusters than $\hat{K}^i$. We simply remove the cluster with the smallest weight. An alternative is the following merging strategy: take the cluster with the smallest weight, and merge into its nearest neighbor. The merged distribution has weight equal to the sum of the weights of the clusters merged, and has mean and variances equal to the effective mean and variance of the two. More precisely, for two clusters with properties $[w_1, \bar{\mu}_1, \bar{\sigma}_1]$ and $[w_2, \bar{\mu}_2, \bar{\sigma}_2]$ (note that we denote the variance by $\sigma$), the merged cluster has properties

$$[w_1 + w_2, \mu_m, \sigma_m],$$

where $\mu_m = w_1\bar{\mu}_1 + w_2\bar{\mu}_2$ and $\sigma_m = w_1(\bar{\sigma}_1 + \bar{\mu}_1^2) + w_2(\bar{\sigma}_2 + \bar{\mu}_2^2) - \mu_m^2$. Our results show that the removal strategy produced better AIC than the merging strategy.

**Splitting the largest cluster.**  This happens whenever $\varphi^i$ has fewer clusters than $\hat{K}^i$. We take the cluster with the largest weight and split it into two clusters. The new clusters are centered at equal distance from and on two sides of the original cluster, and their variance is a scaled version of the variance of the original cluster. Specifically, denote the properties

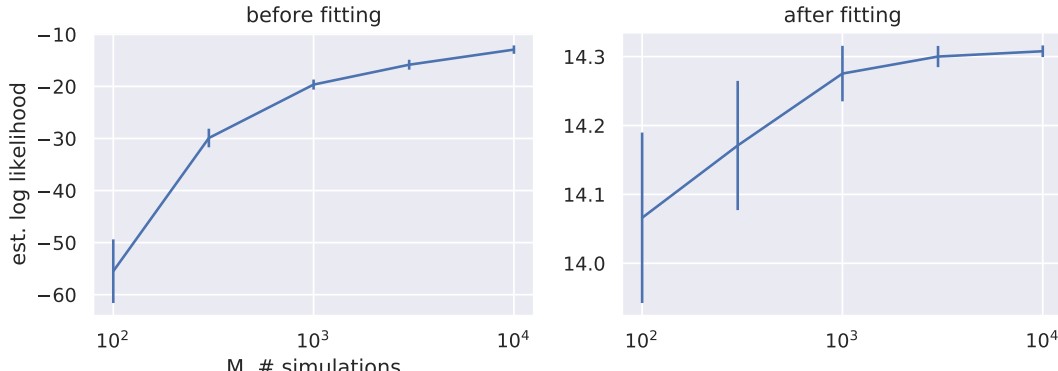

Figure 10: Estimated log-likelihood of data for one participant given different numbers of simulations. We show the mean and standard deviations of the estimate based on 30 runs for each value of $M$.

of this cluster to be split by $[w, m, \sigma]$, we set the cluster properties of the new clusters to be

$$\left[ \frac{w}{2}, \ m - b_\sigma \sqrt{\sigma}, \ s_\sigma \sigma \right], \quad \left[ \frac{w}{2}, \ m + b_\sigma \sqrt{\sigma}, \ s_\sigma \sigma \right]. \tag{11}$$

where $b_\sigma > 0$ and $0 \leq s_\sigma \leq 1$ are free parameters to be optimised. We also tested a version of this strategy where these two parameters are fixed at $b_\sigma = \frac{\sqrt{3}}{2}$ and $s_\sigma = 1/4$, but this gave worse AIC.

**Noise processes.** The noise processes defined in (6) are given by

$$p_w(\mathbf{w}; \hat{\mathbf{n}}) = \mathrm{Dirichlet}(\mathbf{w}; c(\hat{\mathbf{n}} + 1)), \tag{12}$$

$$p_\mu(\boldsymbol{\mu}^r; \hat{\boldsymbol{\mu}}) = \prod_{k=1}^{\hat{K}} \mathcal{N}(\mu_k^r; \hat{\mu}_k, \sigma_\mu), \tag{13}$$

$$p_\mu(\boldsymbol{\sigma}^r; \hat{\boldsymbol{\sigma}}) = \prod_{k=1}^{\hat{K}} \log \mathcal{N}(\sigma_k^r; \hat{\sigma}_k, \sigma_v). \tag{14}$$

where $\hat{K}$ is implied from $\hat{\varphi}$. The addition by 1 in (12) ensures that the mode of the Dirichlet distribution is equal to $w \propto \hat{n}$. The scaling parameter $c$ changes the confidence. The variances $\sigma_\mu$ and $\sigma_\nu$ are free parameters.

### B.3 Fitting algorithm

The MC estimator for the likelihood in (7) is unbiased. However, the corresponding log-likelihood estimator, adopted in practice for numerical stability, is only consistent and produces a nonzero bias for a finite number of simulations $M$. This is due to Jensen's inequality. Suppose each simulation provides a log-likelihood estimate of $\ell_i$ conditioned on some latent variables (such as $\mathbf{z}_T^i$ and $\hat{K}^i$). Let the true conditional likelihood be $X$ so that $e^\ell_i \sim X$. then the estimated marginal log-likelihood is

$$\mathbb{E}\left[ \log\left( \frac{1}{M} \sum_{i=1}^{M} e^{\ell_i} \right) \right] \leq \log\left( \mathbb{E}\left[ \frac{1}{M} \sum_{i=1}^{M} e^{\ell_i} \right] \right) = \log(\mathbb{E}[X]) \tag{15}$$

Note that $\mathbb{E}[X]$ is the marginal likelihood. Fortunately, this bias is downwards, meaning that the expected MC estimate provides a lower bound on the true log-likelihood Still, we must use a large $M$ during training and evaluation, as this reduces the variance of the empirical average in (15) and thus lowers the bias. We show in Fig. 10 the dependence of the estimated log-likelihood for different numbers of simulations, using data from a randomly

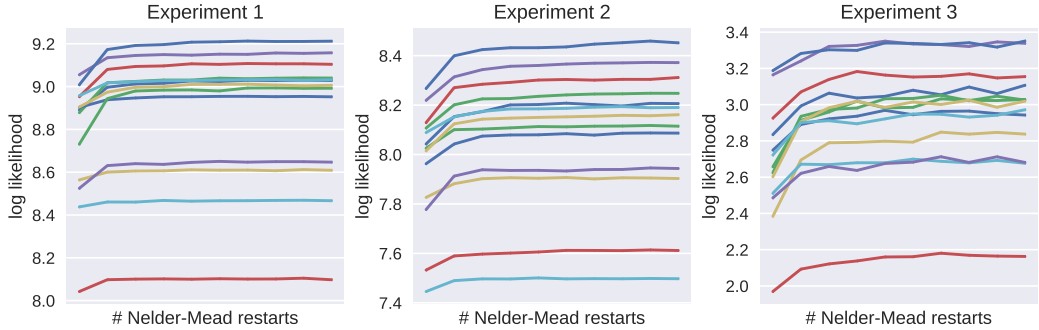

Figure 11: Log-likelihood of different DEFs in ablation studies, including the economical and CRP-GMM DEFs.

picked participant in Experiment 3. When the model is untrained (left panel), the bias does not completely go away even at $M = 10\,000$, but the variance is much smaller than using fewer $M$'s. When the model is well-trained, both the bias and variance of the estimated log-likelihood are substantially reduced (right panel). Thus, the likelihood estimates become more reliable as the DEF fits the data better.

Thanks to the small estimation variance, we are able to fit the parameters of the DEFs using a gradient-free optimization routine Nelder-Mead implemented in the NLOpt package [2]. During training, we estimate the log-likelihood using $M = 10\,000$ parallel simulations on NVIDIA GTX 1080 and A100 GPUs. The latter GPU provides a likelihood estimate within 3 seconds for Experiment 2, and 5 seconds for Experiments 1 and 3. The Nelder-Mead routine typically converges to a 0.001 relative precision on parameters within 300 iterations.

We restart Nelder-Mead 10 times with parameters found from the previous optimization, as this avoids early convergence of Nelder-Mead. To further avoid local optima, we repeat this whole procedure (with 10 restarts) 10 times with different random seeds. We can then check if our algorithm has found a good solution by taking the maximum likelihood found across the 10 repeats. If this solution is reliable, then we should observe that this maximum is stable across the 10 restarts. We then take the best solution among the 10 random seeds at the last repeat as the fitted DEF parameters.

Fig. 11 shows the best (among seeds) log-likelihood across 10 restarts, averaged over all participants, for each DEF setup (see Appendix B.5.1). During the first repeat, the log-likelihood increases and stabilizes. Because we do not keep the best log-likelihood across the restarts, we see small fluctuations across the restarts. We see a very slow increase of log-likelihood for some DEFs in Experiment 3, but the relative rank of different DEFs is mostly preserved.

### B.3.1    Fitting the batch DEF

For the Batch DEF, each simulation maintains $K_{\max}$ submodels before the comparison in (10). Because the cluster weights posterior can be computed during variational inference, unlike in the CRP where we used particles, we do not need as many simulations, and so we run $M = 100$ simulations of each of the $K_{\max}$ submodels. The output of the Batch rational component is then fed into the same aleatoric component for a fair comparison between the ICPs.

### B.4    Particles versus MC simulations

There is an important distinction between the number of particles and the number of MC simulations. In this work, we assume that the participant uses a single particle for inferring the cluster assignments, but this is not observed from the experimenter's viewpoint. The reported internal construct may be associated with one out of many possible cluster

---

[2] https://github.com/stevengj/nlopt

Table 2: Table of all model parameters.

| Parameter name | Symbol | DEF component | Equation | Log-space? | In default? | Notes |
|---|---|---|---|---|---|---|
| Expansion rate | $\alpha_0$ | rational | (1) | Y | Y | |
| Decay rate | $r$ | rational | (1) | N | N | optimized in economical, fixed to 0 in exchangeable |
| Cluster count distortion | $\beta$ | rational | (1) | Y | N | optimized in economical, fixed to 1 in exchangeable |
| Prior mean | $\mu_0$ | rational | (2) | - | - | fixed to zero, not fitted |
| Pseudocount of prior mean | $\lambda_0$ | rational | (2) | Y | Y | |
| Prior variance | $\sigma_0$ | rational | (2) | Y | Y | |
| Pseudocount of prior variance | $a_0$ | aleatoric | (2) | Y | Y | |
| Slack on cluster count | $\epsilon$ | aleatoric | (5) | Y | Y | |
| Concentration scaling in reporting cluster weight | $c$ | aleatoric | (6), (12) | Y | Y | |
| Gaussian noise variance in reporting cluster mean | $\sigma_v$ | aleatoric | (6), (13) | Y | Y | |
| Gaussian noise variance in reporting cluster log variance | $\sigma_m$ | aleatoric | (6), (14) | Y | Y | |
| Visual noise variance | $\sigma_n$ | - | (7) | Y | Y | |
| Poisson prior mean | $\overline{K}$ | batch rational | (9) | Y | N | only in the baseline batch ICP |
| Weight on model size penalty | $\gamma$ | batch rational | (10) | Y | N | only in the baseline batch ICP |
| Cluster weight prior | - | batch rational | - | Y | N | only in the baseline batch ICP |

assignments. As such, we run many simulations of the single-particle particle filter. The number of MC simulations is $M$, and each simulation is indexed by $i$. Since we stick to the single-particle assumption throughout the paper, we do not need an additional index for the number of particles within each MC simulation.

Future work may consider multiple particles. In this case, each MC simulation will contain multiple sequences of cluster assignments. Maintaining multiple cluster assignment sequences allows re-assignment or re-weighting of the observations at each time step, achieving some retrospective correction. However, it is unknown how the participant makes a decision on reporting the cluster properties from multiple samples, especially when the number of clusters do not agree across different particles. We thus leave multiple-particle models for future work.

## B.5   Additional modeling results

After training the DEFs, we evaluate the log-likelihood using a large number of simulations: $10^6$ for the sequential DEFs in our main results, and $10^5$ for the baseline batch DEF.

### B.5.1   Ablation studies on DEF

Our initial experiment design used the exchangeable DEF (with a CRP-GMM rational component) as a reference model. We explored the model space by making small modifications to this DEF motivated by resource constraints and other heuristics. Here, we show that the two modifications in the economical DEF, namely the **Decay** in expansion rate $\alpha_t$ and **Distort**ion in cluster count $n_{t,k}$ produce reliable improvements over the reference exchangeable DEF. The DEFs resulting from other modifications either did not produce reliable improvements or were insignificant compared to the reference model. All model parameters are listed in Table 2. We introduce the modifications below.

**Decay**: adding a model-size dependent decay rate $r$, as in (1).

**Distort**: adding an exponential transformation to the size of the clusters, as in (1).

**Fixed Splitting**: when $\hat{K}^i < K^i$ and splitting a cluster, the parameters of the splitting are fixed with $b_\sigma = \sqrt{3}/2$ and $s_\sigma = 1/4$

**Merge Cluster**: when $\hat{K}^i < K^i$, instead of removing a cluster, the function $f(\varphi, \hat{K})$ merge the smallest cluster into the cluster with closest mean. The cluster properties of the new component is computed by moment matching, as detailed in Appendix B.2.

**Local MAP**: instead of sampling the cluster assignment according to (4), take the cluster with largest posterior probability. This is the local MAP procedure described in [15].

**Constant Variance**: instead of updating the variance of each cluster according to (3), the cluster mean is fixed to the trainable parameter $\sigma_0$

**Fixed Mean**: instead of updating the mean of each cluster according to (3), the mean is fixed at the first observation assigned to the cluster. This checks if participants are able to update the mean or simply remember the first observations assigned to the clusters.

**Fixed Mean Confidence**: instead of fitting $\lambda_0$ as a parameter, we fix it at $\lambda_0 = 0.01$

**No Visual Noise**: do not add Gaussian noise to the observations.

**No Distribution Prior**: when computing the per-step assignment posterior in (4), the likelihood is a Student's $t$-distribution obtained from having a conjugate prior over the Gaussian distribution parameters. Instead of computing this likelihood using the $t$-distribution, this modification computes this likelihood by a Gaussian with mean $\mu_{t,k}$ and variance $\sigma_{t,k}$.

**No Counting Prior**: in (1), instead of using the $n_{t,k}$ maintained for each cluster, use the average cluster size.

The results of these DEFs, averaged over participants, are shown in **??**. Clearly, only Decay and Distort produced reliable improvement on AICs compared to the CRP-GMM DEF across all Experiments. All other modifications that deviate from a rational approximation

of the Bayes rule (e.g. Fixed Mean, No Counting Prior, etc.) resulted in significantly worse fit to the reported internal constructs from our participants. We also see that removing visual noise is detrimental to the quality of the fit.

The stability of the likelihood approximated from (7) is shown in Fig. 11. All DEFs fitting converged well as the number of restarts increases, except for a few worse DEFs in Experiment 3.

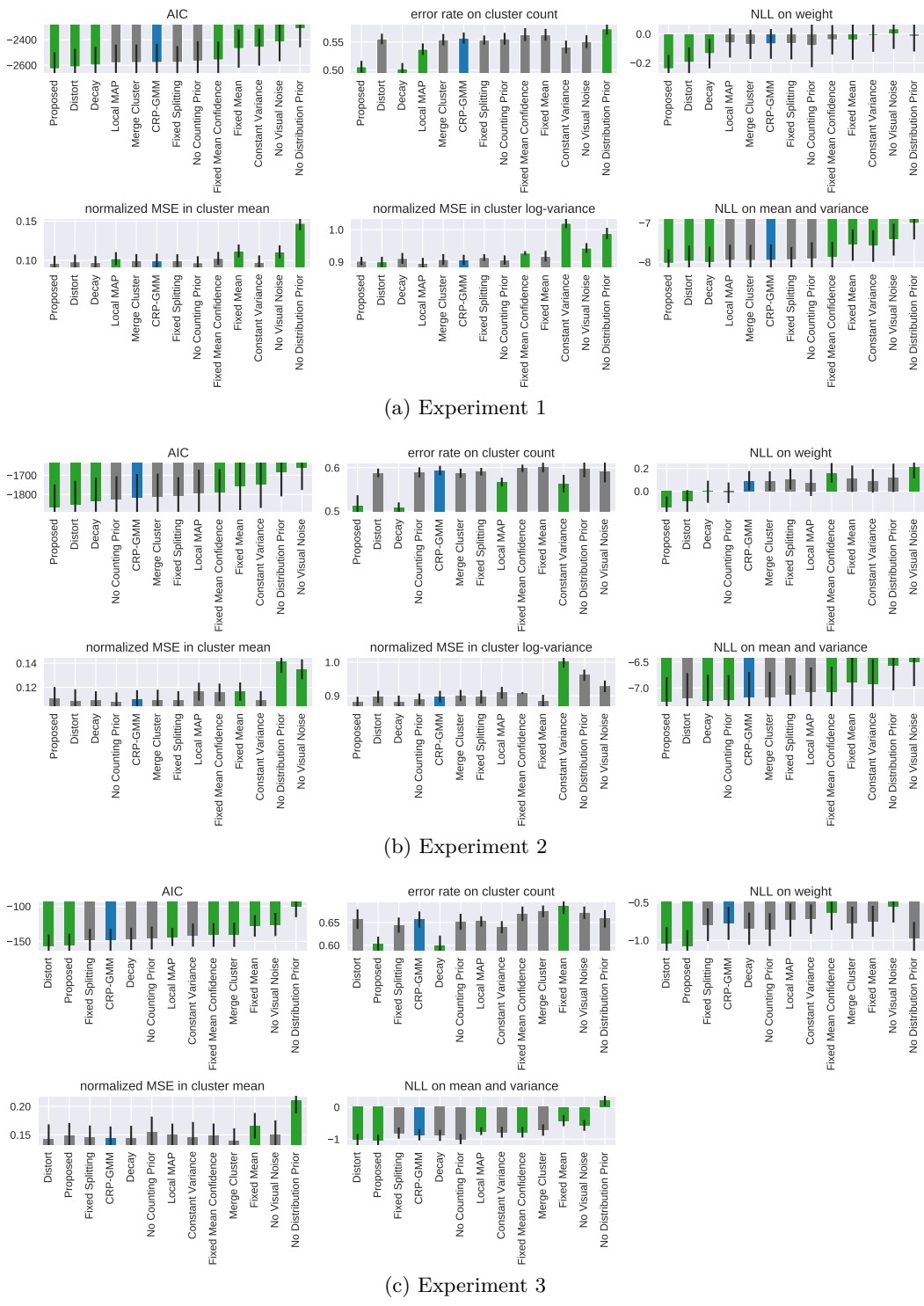

(a) Experiment 1

(b) Experiment 2

(c) Experiment 3

Figure 12: Model ablation comparisons for all Experiments. Lower values are better. Error bars are 1 sems. Blue bar indicates the reference DEF with CRP-GMM as the rational component. Green bars indicate significant differences to the reference model (Wilcoxon signed-rank test, $p<0.05$), and grey bars indicate insignificant comparison. The Proposed (Distort + Decay) is the economical ICP.

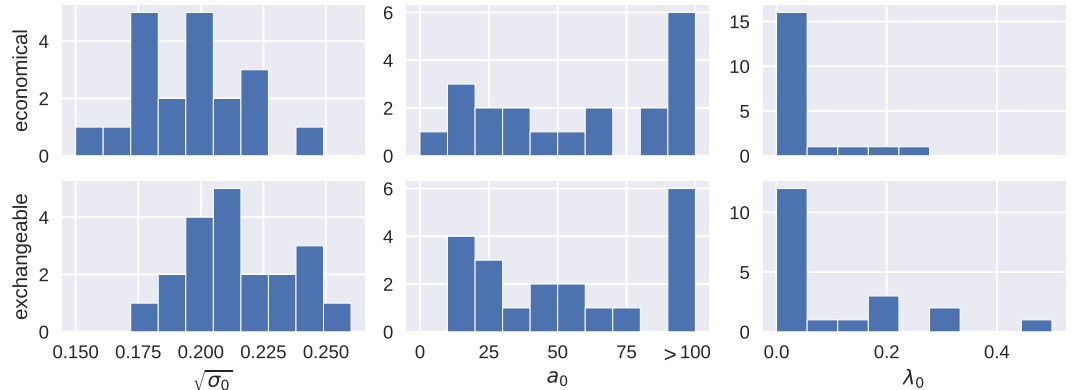

Figure 13: Distribution of fitted ICP parameters using data in Experiment 2.

### B.5.2   Strong prior on cluster width

We found in most fitted ICPs that the prior cluster standard deviation is small (see Fig. 5 and Fig. 13), and the confidence indicated by its pseudocount $a_0$ is much greater than zero. This high confidence contrasts with the low pseudocount associated with the prior cluster mean $\lambda_0$. This means that participants could learn the cluster mean mostly driven by the observations while failing to adapt to the cluster uncertainty. This contributes to the large number of clusters seen when there is a single Gaussian in the true data distribution. We noted that Gershman and Niv [35] also used a relatively high $a_0 = 10$ parameter for their task.

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
