# OpenReview forum: "Bounded rationality in structured  density estimation"
_NeurIPS.cc/2023/Conference — NeurIPS 2023 poster_

### Official Review · Reviewer_ziH7 · 2023-07-04

**Soundness:** 2 fair
**Presentation:** 2 fair
**Contribution:** 2 fair
**Rating:** 5
**Confidence:** 2

**Summary:**

The author built a new model to explain the human mental model of density estimation given sequentially observed data points. The model consists of a “Rational component” that generalizes the Chinese restaurant process. And an “Aleatoric component” that adds some error term. The model is fitted on real experimental data and showed superior performance compared to baseline.


**Strengths:**

The real human experimental data application looks interesting and novel.

**Weaknesses:**

There are some parts of the paper that are unclear to me. See the questions section below.

**Questions:**

Experiment:

I wonder how much of the over-estimation of cluster sizes is due to the small size of samples we generated? Presumably, if the ground truth number of clusters is huge, the human being will instead under-estimate the number of clusters?

Rational component:

It doesn’t seem too clear to me why is the process “economical”. Could you clarify this? My best understanding of the author’s argument is that when the process is exchangeable like a CRP, it involves memorizing all the points which is cognitively implausible. But I think as long as we are doing bayesian inference and getting posterior, it implies memorizing even if things aren’t exchangeable. Another interpretation is that the author is suggesting that participants are not doing a fully bayesian posterior estimation, but some step-wise procedure.

I wonder if it would be useful to give some discussion of comparing economical ICP with say pitman-yor process? (e.g. some more literature review of extensions of CRP?)

Aleatoric component:

In general, I am a bit confused by this part. Is it just a technical point to make MLE feasible, or is it trying to model humans making mistakes?

Line 194: It isn’t clear to me why “the dimensionality of the report is variable” is a challenge that needed to be solved with an aleatoric component. Could you clarify this? I thought nonparametric Bayes prior like CRP already adapts to variable dimensions well. Are you saying that the aleatoric component is definitely needed as otherwise the model cannot be fit at all? Or just that it makes things more numerically stable or realistic?

Formula (5), seems a bit too simplified, is it computationally trickier to set something that gives probability based on how close K_hat is to K_i? I understand that the author suggests that we can pick whatever slack functions but I feel this particular choice somehow seems too naive.

Figure 5(a): It's unclear to me the major differences between economic vs exchangeable methods?

General:

I wonder if there should be something that models human forgetting previous examples and upweighting examples near the end. As far as I can tell that's not in the current framework.

Typo:
Line 148: n_t,k number of “clusters” assigned to cluster k. clusters should be samples?


**Limitations:**

The authors discussed about the limitations of their work in the last paragraph. I wonder if they can provide more discussions on the possible future work to address these limitations?

---

> ### Author Rebuttal · Authors · 2023-08-09
>
> We thank the reviewer for his/her insightful feedback and questions.
>
> Reply to ***How much of the overestimation of cluster sizes is due to the small size of samples?***
>
> Unlikely. If the overestimation of # clusters were due to a small sample size, then we would've observed a lower # clusters in the 70-sample vs 10-sample conditions. This is not the case as we have shown in Figure 1D: the reported # clusters is higher in the 70-sample condition.
>
> Reply to ***If the ground truth number of clusters is huge, the human being will instead underestimate the number of clusters?***
>
> Figure 1D suggests so: humans underestimate the number of clusters when the true number of clusters is 4. Our model also predicts so, because new clusters are less likely to form when $K_t$ is already large.
>
> Reply to ***Why is the model economical?***
>
> We thank the R for providing their interpretation of our model. The model is named economical because it is more conservative in introducing new clusters than the conventional CRP. This makes the cluster assignment prior order-dependent, as illustrated in Appendix B.1.1. We assume that this constraint is mainly due to limited memory capacity.
>
> As the R pointed out, we do not perform full Bayesian posterior estimation. However, this feature is shared between the economical ICP and exchangeable ICP, so this is not the main reason for calling the model we proposed as being economical.
>
> Reply to ***Comparing economical ICP with other extensions of CRP***
>
> The suggested Pitman-Yor process (PYP) has a discounting parameter to control how fast new clusters are added. At each time $t$,
> $$
> P(z_{t+1}=k|\mathbb{z}_{t}) := \frac{\alpha + n_k b}{\alpha+t}, \quad  k = K_t + 1
> $$
>
> or
>
> $$
> P(z_{t+1}=k|\mathbb{z}_{t}) := \frac{t - b}{\alpha+t}, \quad  k \le K_t
> $$
>
> To slow down cluster creation, we must make $b<0$. As $K_t$ grows, the probability of expansion becomes negative, posing a more challenging constrained optimization problem. The economical ICP modulates only the expansion rate through a monotonic decreasing function of $K_t$, so it is more flexible, interpretable, and easier to optimize.
>
> We are also aware of other priors, such as the sticky CRP [*] used for memory updating [**]. However, we did not find this parametrization to improve the model fit to data in our earlier experiments. We will discuss the PYP, the sticky CRP, and other CRP-like variants in a revision.
>
> [*] Fox E B, Sudderth E B, Jordan M I, et al. A sticky HDP-HMM with application to speaker diarization. The Annals of Applied Statistics, 2011
> [**] Gershman S J, Radulescu A, Norman K A, et al. Statistical computations underlying the dynamics of memory updating. PLoS computational biology, 2014
>
> Reply to ***Purpose of the aleatoric component***
>
> The aleatoric component can be regarded as modeling human mistakes. Note that cognitive and behavioral noises are observed in many cognitive science studies, so the including the noise component is a standard procedure in modeling. See the ***Intuition and motivation for the aleatoric component (8bgA, ziH7)*** section in our general response for a detailed reply. We are happy to provide further clarification if the R could elaborate on their confusion.
>
> Reply to ***Variable dimensionality***
>
> We clarify that the challenge is not how to produce variable-dimensional predictions, but rather how to measure the likelihood of the model when it produces such predictions. For each trial, different simulations of the DEF produce different numbers of clusters, but the reported cluster parameters have fixed dimensionality for each trial. To our knowledge, modeling variable-dimensional data is not well explored in cognitive science, so we regard this as a challenge for modeling.
>
> Reply to ***Is the aleatoric component definitely needed?***
>
> The aleatoric component is definitely needed, because the rational component alone could not capture the distribution of the reported data. Please see our general response for a detailed reply.
>
> Reply to ***Equation (5) may be too simple***
>
> One could also choose other conditional distributions over K for (5); however, we want to encourage the expected number of correctly predicted K to be high, rather than to encourage e.g. the expected $\ell-1$ or $\ell-2$ errors to be small, such as would result from a Laplace- or Gaussian-like slack distribution.
>
> ***Difference between economical and exchangeable in Figure 5(a)***
>
> This panel only provides example predictions from the two models. The major difference is in the relative weights of the clusters. The economical model has a distortion on the reported weights, and the predicted weight values are pulled to be closer to each other (more homogeneous).
>
> ***Modeling human forgetting previous examples and upweighting examples near the end***
>
> These effects are known as recency effects in memory literature. We tested both the recency and primacy effects (up-weighting very first examples) by adding a weighting function (decaying for the primacy effect, increasing for the recency effect) to the cluster assignments. However, we did not find this to improve the model fit. We also tried to directly look for traits of primacy and recency effects in the data but did not find any significant results.
>
> ***More discussions on limitations***
>
> We will incorporate more limitations mentioned by the reviewers in a revision. In particular, we will address limitations in our experimental design, such as not asking for intermediate reports from participants. Finding a suitable way to probe for this without inducing additional cognitive effects, such as confirmation bias and forced certainty, is an important future direction. In terms of modelling, we will point out other more advanced versions of CRP, such as the repulsive CRP [\*] that may help capture the non-overlapping clusters reported by participants.
>
> [\*] Quinlan J J, Quintana F A, Page G L. On a class of repulsive mixture models[J]. Test, 2021

---

> > ### Comment · Reviewer_ziH7 · 2023-08-18
> >
> > Thank the authors for their detailed response! They have addressed my questions, so I'm willing to increase my score to 5.
> >
> > Regarding to Comparing economical ICP with other extensions of CRP: here is another CRP-like variant: Zhou, D., Gao, Y. and Paninski, L. Disentangled sticky hierarchical dirichlet process hidden markov model. ECML PKDD 2020.

---

### Official Review · Reviewer_8bgA · 2023-07-06

**Soundness:** 3 good
**Presentation:** 2 fair
**Contribution:** 3 good
**Rating:** 7
**Confidence:** 3

**Summary:**

The authors describe a density estimation task with humans wherein participants were asked to identify parameters of an unknown distribution from presented data. While participants did a better job of estimating the overall density with more samples, the authors observed a large error in the reported number of clusters in the underlying distributions (and that increased with sample size but typically fell in the range 2-3).  The participants also tended to estimate the clusters as non-overlapping.  They modeled this process with a non-parametric mixture model.   This model consists of a “rational” component and an “aleatoric” component.  The rational component uses for cluster assignments an extension of the Dirichlet process that reduces the relative probability of adding new clusters as new points are added and modulates the relative probability of assignment to an existing cluster with a form of divisive normalization.  Unlike the regular Dirichlet process, this process is not in general exchangeable, a property the authors argue is cognitively implausible and unnecessary.  The aleatoric component adds “structured noise” in order to compute an approximation to the marginal likelihood.  They compare their model against both an exchangeable version and a baseline “batch” learning prior.

**Strengths:**

This is an interesting paper about the important problem of how people form internal models from observations.  The authors use an interesting approach with a non-parametric prior and provide a fitting routine that uses state-of-the-art tools for efficiency.  The experiments are clearly described and the model is (mostly) well presented.  The evidence that humans do not form models based on exchangeability is an interesting result.

**Weaknesses:**

Overall, the presentation is excellent, but the description and motivation of the aleatoric component is lacking, to me.  Adding some intuition about why you made the choices you made for the model would be helpful.  The “slack” probability seems particularly arbitrary to me, as does the noise added in equation 6.  Why are these necessary?  You mention in the appendix that “the DEF must place non-zero likelihood to mixture models with all possible numbers of clusters” which makes sense because many potential clusterings could result in the same reported number of clusters, but then you essentially remove from the estimation (equation 7) any sample in which the number of clusters does not match what is reported (line 208).  The “slack” probability seems to carry all of the weight of this possibility.  It is likely I’ve misunderstood something, but I think this section could be clearer.

The results from Figure 1 raise some interesting questions about the need for this particular model.  I can’t tell since they are bar plots, but it makes me wonder how consistent the cluster number estimate is and what its dependence is on the number of data points.  Moreover, can the data support a simpler psychological heuristic that assigns a number of clusters based on the number of observed samples, particularly given that subjects tend to estimate non-overlapping clusters?  It is not clear to me whether the baseline model accounts for this possibility.  I’m imagining a heuristic where people divide the screen into chunks and assign clusters that way, making finer divisions with more and more data points.  Is it clear that this is not what is happening?  Perhaps this is an ignorant question, but it seems to me that the estimation should be driven by some important visual priors.


Minor:

Line 129-130:  detailed mechanics are [not?] directly obvious…
Line 150:  For r > 0, new clusters are… (not r< 0).
Line 206:  There [are] \hat{K}^i clusters…

**Questions:**

What is the spread of data in Figure 1, particularly panels B and D?  Are participants consistent in the number of clusters assigned?  Is there a potentially simple relationship between number of clusters and sample size?  Are there simpler null models to compare against that make some relatively add hoc cluster assignments of this nature?  Maybe the baseline model behaves this way, but it is not clear to me that that is the case.

K_max from the aleatoric component is across participants, which means the posterior takes into account observations from other participants.  Is this a practical concern?  Can you replace K_max with some large but not unwieldy number and get similar results (I assume so)?

This is beyond the scope, I think, but is there a way to get running estimates of the distribution over the course of the experiment?  Your experiments don’t directly have this information, but can you get a clue to how people update models of distributions by comparing the experiments with different sample sizes?  This seems a natural question given the result that non-exchangeability seems to be important in modeling the human behavior.

**Limitations:**

The authors do not discuss societal impact nor directly discuss limitations.

---

> ### Author Rebuttal · Authors · 2023-08-09
>
> We thank the reviewer for helpful comments and clarifying questions.
>
> There are many important choices in the aleatoric component, so we provide an elaborated explanation here. We will revise the manuscript and the supplemental material to add these descriptions.
>
> ***Reply to "Intuition and motivation for the aleatoric component."***
>
> It is a cognitively plausible component that helps us fit the model. We regard the rational component as modeling the density estimation process while stimuli are presented, and the aleatoric component as modeling mostly the behavioral noise while participants report the cluster parameter. Please see ***Intuition and motivation for the aleatoric component*** section in our general response for a detailed reply.
>
> ***Reply to "Why is the chosen slack distribution (Eqn 5) necessary?"***
>
> The noise model on $K$ must assign none zero probabilities to all possible model sizes $K$ within a reasonable range, while encouraging the inferred $K^i$ from the ICP to be sampled most of the time, as it depends on the stimuli. This thus motivates the discrete K slack distribution in (5). Crucially, the simulations with incorrect $K$ are not simply removed but instead contribute zero conditional likelihood to the summation in (7). Since $M$ is fixed regardless of how many simulations yield the correct K, the ICP must produce as many as possible simulations with the correct $K$ to achieve a high likelihood. Note that the value of $\epsilon$ does not enter the likelihood computation explicitly, but only through sampling of the slack distribution. One could also choose other conditional distributions over K for (5); however, we want to encourage the expected number of correctly predicted K to be high, rather than to encourage e.g. the expected $\ell-1$ or $\ell-2$ errors to be small, such as would result from a Laplace- or Gaussian-like slack distribution.
>
> ***Reply to "Why is the chosen cluster parameter distribution (6) necessary?"***
>
> The noise model for cluster parameters must preserve the strong dependences between
> 1. the cluster parameters and reported K,
> 2. between the cluster parameters themselves.
> The latter comes from the permutation invariance of the clusters in defining the density function. These considerations motivated the seemingly complicated noise model and cluster adjustment functions in (6). Without the cluster adjustment function, the cluster parameters become decoupled from the slacked $\hat{K}^i$. The summation over the permutations ensures that the likelihood is insensitive to the order in which the distribution parameters appear in the ordered vector representation.
>
> ***Reply to "Other heuristics where the number of clusters depends on the sample size"***
>
> It is possible that humans build a different prior for each sample size. However, in experimental settings where the sample size varies across trials and is thus unknown before the sequential presentation of samples (e.g. in Experiments 1 & 3), this heuristic would be incompatible with online learning. That is, agents using such heuristics need to memorize all the samples before inference, which is less cognitively plausible. For Experiment 1, we implemented a sample size-dependent prior (Poisson given each sample size) under the batch learning ICP. This gave an insignificant improvement of $\Delta AIC=5$ (p=0.59) to batch baseline, but still far worse than the exchangeable or economical DEF. In experimental settings where the sample size is fixed (e.g., Experiment 2), this heuristic would reduce to the batch baseline model we tested in the paper and found to be inferior to the exchangeable DEF and the economical DEF.
>
> ***Reply to "Boundary-based heuristic model"***
>
> We thank the R for the suggested intuitive model, but we do not fully understand the suggested procedure of “making finer divisions with more and more data points”. It is possible that participants store not the clusters' means and widths, but the boundaries. However, a sensible boundary must rely on memorizing all the previous stimuli so that the divider does not cross densely populated regions. This renders this procedure less cognitively plausible. Memorizing densely populated regions is then tantamount to having a cluster-based representation, such as the GMM-based ICP we use.
>
> ***Reply to "What is the spread of data in Figure 1, particularly panels B and D?"***
>
> To visualize the spread, we replot Figure 1B and 1D in response PDF (Panels A & B), with individual participants’ data points denoted by grey dots linked by grey lines.
>
> ***Reply to "Are there simpler null models to compare against?"***
>
> Please see the general response. We also ran two additional batch ICPs using priors suggested by other reviewers. These are still significantly worse than the sequential DEFs.
>
> ***Reply to "K_max from the aleatoric component is across participants"***
>
> This is not a practical concern, because the amount of information provided by this $K_{max}$ is very small: only a tiny fraction of the reported $K$ reach $K_{max}$. Even if we increase $K_{max}$, then the K-slack model assigns a nonzero probability ($\epsilon$) to a value that never appears in the dataset, so this will lower the likelihood. In this sense, $K_{max}$ is the maximum-likelihood solution of the maximum predicted $K$ for the slack model. Computationally, a $K$ too large induces a much larger number of clusters, which makes the sum in (6) over permutations impractical to compute.
>
> ***Reply to "Is there a way to get running estimates of the distribution over the course of the experiment?"***
>
> We considered this during our experimental design and decided not to go for this choice. Asking our participants to report on the fly may create confirmation biases or force them to commit to a distribution that is originally uncertain. Instead, we designed the 10/70 sample-size comparison without an intermediate report to test for the effect of sample size on the reported distribution.

---

> > ### Comment · Reviewer_8bgA · 2023-08-20
> >
> > Thank you for the detailed response.
> >
> > I don’t agree that a boundary based estimated driven by visual priors would require memorization of all previous data.  At worst perhaps one needs to memorize extremal points.  However, if one permits some degree of error (as seems reasonable in context) then ascribing clusters to approximate visually bounded regions still seems to make some sense.  I do agree however, that your model is a good approach to this kind of approximate clustering.  How and whether it relates to visually driven spatial priors is a question that would be unfair to ask you to address, even though interesting.
> >
> > Thank you for the new plots and the new analyses with other models.  I am adjusting my score.

---

### Official Review · Reviewer_ti72 · 2023-07-06

**Soundness:** 3 good
**Presentation:** 3 good
**Contribution:** 3 good
**Rating:** 7
**Confidence:** 4

**Summary:**

This paper presents a visual density estimation task for human subjects based on sequential data from gaussian mixture models, complete with experimental data analyzed through a bounded-rational model of behavior. The paper reports data on the quality of the density reported by subjects, claiming that it roughly matches the true distribution's first three moments while highlighting some systematic mismatch in the number of reported mixture components. A variant of the task is performed using a different modality (numerical data instead of visual stimuli). After the task and the data is presented, the paper develops a bounded rational model for behavior. This model is composed by a generalization of the chinese restaurant process plus a sophisticated noise component. The main difference from the CRP is that this model implements in general an "economical" density estimation process where the likelihood of adding new clusters decreases faster than in the CRP. The behavioral model comes with an inference scheme implemented in pytorch, which allows for gpu acceleration. This model is applied to the data, showing that the "economical" version explains the data better than simpler alternatives (regular CRP or batch learning where cluster assignment is not sequential). This is argued to support the idea that subjects are affected by constraints on memory capacity.


-----
(Scores edited following the author's rebuttal)

**Strengths:**

- The paper contains a large amount of novel work, including a behavioral task, a bounded rational observer, and advanced techniques for inferring the parameters of the observer and comparing it with alternatives explanations of empirical data.

- The task is ambitious in that it has a high-dimensional report.

- The bounded rational observer is nontrivial but well explained, despite the limited space. From a statistical standpoint, this model and its efficient implementation are valuable.


**Weaknesses:**

1. The statements that the subjects's density estimates are reasonably good do not seem sufficiently supported by the data. Line 86: "the reported densities tracked the ﬁrst three moments reasonably well", but figure 1 does not show the data (the values of the true and reported moments), only the estimated correlation between them. Line 105: the density estimation quality is "reasonably good" in experiment 3, but figure 7 does not show any data on this - it only shows that the economical DEF reproduces certain patterns in subject behavior.

2. As pointed out in the paper (lines 291-292, "the prediction errors on the number of clusters and cluster variance are still large"), the economical internal construct prior (ICP) model is simply not very good at describing the empirical behavioral data. This is not necessarily a problem by itself, but it makes it difficult to draw any conclusions about human behavior from the application of the model to this dataset. In particular, the claim that "We provide key experimental and modeling evidence that humans may employ an online and yet ﬁnite (in expectation) model" does not seem adequately supported: if the economical ICP does not describe well subject behavior, it does not matter that it does comparatively better than the CRP or batch learning.



**Questions:**

1. Can you provide more evidence that the quality of the density estimation reports is "reasonable", despite the clear systematic errors in the number of components?

2. Can you clarify why the model provides experimental insight into human behavior, if it does not capture basic features of the behavior like mean and variance of the components? Moreover, how is this mismatch compatible the claim that "The DEF with the economical ICP captures all of the behavioral patterns reported in Section 2, including the inconsistent number of clusters and the low overlap ratio between adjacent clusters"? Figure 5A (referenced in this passage) only shows three examples; I can't find any quantitative or statistical support for this claim. Alternatively, the claims about the insights derived from the application of this model to human behavior should be significantly reduced.

3. Figure captions should describe what's in the figure, not (just) provide comments. For instance, the caption of Figure 1E says "adjacent clusters in the reported distributions did not overlap much", but does not say what is being plotted there. How is the overlap computed? The figures would be clearer if this pattern/tendency in the captions was avoided.

**Limitations:**

The paper does highlight limitations in the results, although (as discussed above) other passages claim conclusions that do not seem supported by the data. Potential negative societal impact is not a concern here.

---

> ### Author Rebuttal · Authors · 2023-08-09
>
> We appreciate the reviewer's suggestions on providing more details about human data and model fits, and more quantitative support for qualitative statements. These suggestions have inspired us to dig deeper into our data and model fits, which has yielded richer results (see response PDF) that we believe are more convincing. Please see detailed responses below as well as the general response to all Rs.
>
>
> ***Reply to "More evidence that the quality of the density estimation reports is reasonable"***
>
> In our previous manuscript, we showed that subjects’ density estimates roughly match the sample distributions in their moments (i.e., mean, sd, and skewness, Figure 1C for Experiment 1) Following the R’s suggestions, we have enhanced these plots of moments for Experiment 1 (panels D&E in response PDF) and added similar plots for Experiment 2 (panel F). In these new plots, we included three layers of data:
> 1. the distribution of responses of individual trials (collapsed across subjects), plotted as the grey heatmap in the background.
> 2. average responses and individual differences across subjects, plotted as the 5 black dots and their error bars (standard error).  To compute the 5 average responses, the values of the sample mean (sd, skewness) are evenly divided into 5 bins, and the trials in each bin are averaged, first separately for each subject and then across subjects.
> 3. the regression lines as before.
>
> Along with these plots of reported moments, we also present the fits of our economical DEF (sub-panels highlighted by red dots and lines). We see that the economical DEF captured not only the report of the sample mean (first row of panels D–F) but also the patterned errors in the report of sample sd and skewness (second and third rows of panels D–F). The PDF includes a few more comparisons between human data and model fits where we visualize much richer patterns in the high-dimensional human data and show how well our model fits these patterns. All of these results suggest a reasonably good fit of the economical DEF to behavioral data.
>
> ***Reply to "The estimated model does not capture data well"***
>
> Apologies for the misunderstanding caused by the original manuscript. To visualize that our model is capable of capturing various aspects of human data, we provided a more thorough illustration of model behavior in the response PDF. The fitted model behaves like the human subjects in both (1) the moments of the reported marginal density and (2) the reported cluster properties. Please see the **How well does the model produce human data? (Ds9y, Xmkb, ti72)** section in the general response for the detailed discussion.
>
> ***Reply to"What we can learn from the model if it does not capture the mean and sd."***
>
> The plots in the response PDF show that the model can capture the mean, sd, and other measures in the human data. Please see the replies above as well as the general response for details.
>
> ***Reply to "Figure 5A only shows three examples and a lack of quantitative and statistical support"***
>
> Figure 5A shows example trials, and the figures showing the overlap ratio and inconsistent report are in Figure 7 of Appendix B.4. Sorry for the slightly confusing cross-referencing.
>
> ***Reply to "Overclaims that are not well supported"***
>
> We will tone down the overly strong statements at the suggested places and elsewhere. We will make sure there are no other statements without data or statistical support.
>
> ***Reply to "Figure caption styles and computation of the overlap ratio"***
>
> We provided an intuitive visualization of the overlap ratio in the figure, but agree that we should be more explicit about the definition.  Overlap ratio is defined as the percentage of a cluster’s area being covered by any other clusters averaged across all clusters in the report.  Such computation of overlap ratio has the advantage of not being affected by the number of clusters. We also tried other measurement of the overlap ratio (e.g., the area under the second highest cluster density curve), which gave similar results. The overlap between clusters is low. We will add details of the computation of the overlap ratio to the revised text.
>
> Adding comments is common in figure captions in cognitive science papers. We will make sure that the comments are clear and faithful to the content, and that all variables/features are clearly defined in the main text. We appreciate the R’s comments.

---

> > ### Comment · Reviewer_ti72 · 2023-08-15
> >
> > Thank you for replying to my questions. The additional analyses/plots go a long way towards addressing my concerns - especially the central one, that there was not enough evidence that the DEF with economical ICP captures well subject behavior. I have increased my scores accordingly.

---

### Official Review · Reviewer_Xmkb · 2023-07-08

**Soundness:** 3 good
**Presentation:** 3 good
**Contribution:** 3 good
**Rating:** 8
**Confidence:** 4

**Summary:**

The authors consider the question: how do humans estimate probability distributions? They study this by performing three experiments where human subjects, where they ask participants to recover Gaussian mixture models after seeing IID samples. They find that while subjects do seem to get closer to the true mixture model as they see more samples, they consistently seem to report about three different components regardless of the ground truth number of components. They speculate that this is due to humans having limited memory.

They then propose a human model consisting of a rational component that performs approximate Bayesian inference which generalizes the standard Chinese Restaurant Process (CRP) and an _aleatoric_ component where humans pick the number of components and then merges and splits clusters until the desired number of components is reached (with some noise). They derive an efficient estimator for the likelihood under this model using Monte Carlo simulation. They then show that their model fits their experimental data much better than the standard CRP. The better performance of their model suggests that it does not seem necessary to keep the human prior exchangeable.

**Strengths:**

* The paper is generally clearly written, and the authors do a good job of motivating their problem and describing their results.
* The authors' structured online density estimation task is clean and straightforward.
* The key experimental observation (that the human subjects seem to consistently estimate around 2 or 3 clusters, regardless of ground truth) is very clearly presented and clearly of great interest to this subfield of human modeling.
* The authors two additional experiments validating the key experimental observation, with different distributions and different domains.
* Their proposed aleatoric component follows naturally from their key observation, and cleanly captures the inconsistent number of clusters reported by human subjects.
* The authors experimentally validate that their proposed model generally performs better than the standard CRP (as well a batch learning baseline) for all three experiments, using five different criteria: AIC, #clusters correct, negative log likelihood, the error in means, and the error in log variances.
 * The authors give the fitted parameters and discuss how to interpret them, which I found quite interesting.

**Weaknesses:**

* All three experiments featured relatively simple Gaussian mixtures -- for example, experiment 1 has 1-4 clusters, experiment 2 has 1-3 clusters, and experiment 3 looks at _unimodal_ Gaussian mixtures/distributions. As a result, it seems hard to distinguish the interpretation where human subjects have limited memory and are using the proposed ICP from the interpretations where human subjects just happen to estimate 2--3 clusters across the board, or where they consistently tend to overestimate the number of clusters for other, unrelated reasons.
* Similarly, the proposed ICP seems quite convoluted (e.g. the authors' likelihood estimator requires Monte Carlo simulation) and I'd be surprised if human subjects were actually using it.
* Despite the complexity of the proposed ICP and the relative simplicity of the distributions being studied, the error bars for fitting human predictions are still relatively large (as shown in Figure 4), suggesting that it does not accurately capture the underlying human behavior.
* I found figure 3 confusing; in particular, while the caption was quite helpful to my understanding, I still don't think I understand the rational or aleatoric component parts of the figure, and I did not find them helpful when I was first reading the paper.
* Similarly, I found the discussion of how the aleatoric and rational components were composed to be initially confusing.

**Questions:**

1. It seems like the combination of the rational and aleatoric components are quite complicated -- have you considered simpler models? For example, instead of doing online expansion, perhaps the ICP just has a fixed prior distribution over cluster count.
2. Similarly, given the complexity of the ICP, I'm curious how the authors interpret the fact that human subjects are able to (apparently) use it.
3. As mentioned in the weaknesses section, I'm curious about how these results generalize to more complicated inference tasks. I understand if the experiments have not been done, but what do the authors expect the results to be if you looked at experiments with more clusters or in more realistic settings?
4. I would've appreciated more discussion about the actual implementation of the estimator -- for example, how reliable is the log-likelihood estimate, and how sensitive is it to the number of MC simulations?
5. Similarly, I would've appreciated more analysis of when the predictions of their proposed ICP fall short of reality (as opposed to only cases where the number of clusters agree with human report).
6. As this is a relatively theoretically oriented paper, this is not strictly necessary. However, I wonder if the authors have seen the tendency for human subjects to pick a fixed number of clusters in other contexts, perhaps due to computational constraints. Do you expect this to have implications in practice? Are there more realistic contexts where this tendency causes systematic deviation from "normative" behavior?

In addition, there is the following typo in the paper:
* Line 205-6: ... until **there** $\hat K^i$ clusters -> ... until **there are** $\hat K^i$ clusters

**Limitations:**

* I commend the authors for explicitly highlighting the relatively poor fit (albeit significantly better than prior models) of their proposed ICP on their experiments. However, I would've appreciated more discussion about why this might happen, and what next steps could be taken to address this issue.
* I think the authors should include more discussion of the limitations and weaknesses I mentioned in the weaknesses section above.

---

> ### Author Rebuttal · Authors · 2023-08-09
>
> We thank the reviewer for the positive and helpful feedback.
>
> ***Reply to "It seems hard to rule out the interpretations where human subjects just estimate 2--3 clusters across the board"***
>
> Behaviorally, we see that the distribution of reported $K$ varies across the number of samples and distribution types (Panels B & J in PDF), so participants are not reporting $K$ randomly. Besides the batch baseline model mentioned in our general response, we also implemented another batch ICP in which the prior over $K$ is a discrete distribution over $K=2$ and $K=3$ (note that the aleatoric component still permits more values of predicted $K$). Compared to the Poisson prior in the batch baseline model, the new prior had a small but insignificant improvement (mean $\Delta$AIC $\approx 20.0$, $p=0.61$), still far worse than the CRP-GMM ICP.
>
>
> ***Reply to "ICP seems too complicated to be implemented by the brain"***
>
> This question may arise from a common confusion between the participant’s view and the experimenter’s view of the ICP. In the former, the participant always uses single particle trajectory $z_1:t$ for cluster assignment. However, these assignments are latent variables hidden away from the experimenter (us), so the experimenter needs to consider and average out all possible latent trajectories that could be generated in the participant’s mind. Only the experimenter needs to average (or marginalize) out this latent variable for modeling purposes. Note that Gershman & Niv in [25] also used a large number of simulations to marginalize out latent variables, although the likelihood function used there is more restricted.
>
> ***Reply to "Large error bars of model predictions"***
>
> Please see the general response, ***How well does the model produce human data? (Ds9y, Xmkb, ti72)***
>
> ***Reply to "Combination of the rational and aleatoric components"***
>
> We will further clarify the functions of the two components in a revision. In short, the rational component is the approximate Bayesian inverse of an ICP, implemented as a particle filter. This component produces raw predictions of the reported distribution parameters, as in most other Bayesian models for human behavior. Now, if there were only a single Gaussian in the data and in the participant’s report, then we can add simple noise models to create a valid likelihood function, such as the Dirichlet and isotropic Gaussians in Eqn (6). However, the number of clusters in the report and prediction varies in our setup, so we need to introduce a more sophisticated and structured noise model, the aleatoric component. Please see our general response for intuitions on the aleatoric component.
>
> ***Reply to "Generalisation to realistic scenarios with more clusters"***
>
> One key prediction of the decaying $\alpha_t$ with $K_t$ is that humans are less likely to create a new cluster when there are already many clusters, but with overwhelming evidence from the observations, a new cluster can still be created if, for example, many clustered samples appear one next to each other and far away from existing clusters. In addition, if memorizing the clusters takes up memory capacity, then we should expect the $\alpha_t$ to decrease faster if the dimensionality of the stimulus $x_t$ is higher since each cluster now requires multiple times more space to store. We will further verify these predictions in future work.
>
> ***Reply to "Quality of the parameter estimation procedure"***
>
> Detailed information about the fitting algorithm is presented in Appendix B.3, where we quantify the variance and bias of the estimated log-likelihood as a function of # MC simulations in Figure 10, and we show the convergence of the parameters in Figure 11. Overall, the amount of variance and bias is very small at the optimal parameters found, although convergence is less ideal for Experiment 3. In addition, we present parameter recovery results in panel K of the response PDF.
>
> ***Reply to "Analysis of when the model fails (e.g. to predict the number of reported clusters)"***
>
> This is a very interesting suggestion. Note that the correlation between the statistics of reported and true distributions in Figure 1 does not exclude predictions with an unmatched number of clusters. It is also expected that the distributions would be more smooth if the predicted $K$ is less than the reported $K$, and vice versa since the cluster width is small *a priori* with high confidence. However, the behavior of the predicted cluster parameters is unspecified and unconstrained by data. Perhaps a less stringent aleatoric component may reveal more interesting predictions for an unmatched number of clusters, but any effects would be tightly linked to the inductive biases of the specific aleatoric component.
>
> ***Reply to "Tendency of humans to pick a fixed number of clusters"***
>
> Please see our response to a similar question raised by **Ds9y—"Implication of the results to more naturalistic scenarios”**.  Ravignani et al (2016) [*] found that after several rounds of laboratory music evolutions, participants’ reported distribution of rhythms became much more clustered than the original uniform distribution. According to Figure 2 in their paper, the number of clusters that emerged from participants’ reports seems to be close to 3. We also speculate that the overestimation of the number of clusters is closely related to the perception of illusory causal relationships in human superstitious thinking [**].
>
> [*] Ravignani, A., Delgado, T., & Kirby, S. (2016). Musical evolution in the lab exhibits rhythmic universals. Nature Human Behaviour, 1: 0007. doi:10.1038/s41562-016-0007.
>
> [**] Matute, H., Blanco, F., Yarritu, I., Díaz-Lago, M., Vadillo, M. A., & Barberia, I. (2015). Illusions of causality: How they bias our everyday thinking and how they could be reduced. Frontiers in Psychology, 6, 1–14.

---

> > ### Comment · Reviewer_Xmkb · 2023-08-15
> > **Thanks for your detailed response.**
> >
> > Thanks for the authors' detailed responses. As you have addressed all of my concerns, I am raising my score to an 8 and my confidence to 4.

---

### Official Review · Reviewer_Ds9y · 2023-07-15

**Soundness:** 2 fair
**Presentation:** 3 good
**Contribution:** 3 good
**Rating:** 7
**Confidence:** 3

**Summary:**

This study investigates how humans infer probability distributions from samples by combining experiments and modeling. The main contributions include a careful characterization of the behavioral tendency to overestimate the number of clusters as well as a modeling framework to identify how this behavior can arise from approximate inference. By fitting parameters from a generalized version of the Chinese restaurant process, the authors conclude that behavior can arise from a myriad of factors including strong prior expectations about the sample variance, undercounting samples, as well as a decaying tendency to form new clusters.

**Strengths:**

Human experiments are well-summarized and clearly document a tendency to overcluster samples. By directly asking participants to report the data-generating distribution, the approach provides a direct readout of the participant's biases that are brought to bear when solving complex probabilistic inference problems.

The paper is well-written and easy to read except for the section introducing the modeling framework where I got lost in the notations for a while. But the illustrations were very helpful.

The modeling approach is both rigorous and appropriate for answering the questions tackled by the study.

**Weaknesses:**

I see two main weaknesses.

First is that the insights gained from modeling seem to be quite minimal and specific to the paradigm used here. I'd like to know whether/how the conclusions might apply more broadly to naturalistic scenarios.

Second, given that the model fits are not that great, it is quite possible that there are alternative models not considered here that explain behavior better. A discussion of alternatives is lacking.



**Questions:**

Since the fitting procedure relies on approximate inference, I'd like to see a validation of the model fits to show that the method is able to recover the true parameters. Is this already published or included somewhere in the supplementary?

Based on the best-fitting parameters, the authors point out a trade-off between two different effects. A decaying tendency to form new clusters (which should lead to underclustering) and a strong prior that underestimates the variance of the gaussians in the mixture (which should lead to overclustering). Could the trade-off simply be a result of over-parameterization? Again, some sort of validation of the model fitting procedure would help here.

On a related note, it wasn't clear to me how the model explains why humans report more clusters when sample size increases from 10 to 70. Shouldn't the decaying $\alpha$ lead to fewer clusters as a function of sample size?

Given that data exchangeability is not a constraint for human inference, why not validate the model directly by running a new experiment where the sequence of samples is presented in a reverse order? I suppose the model will have concrete predictions for such an experiment.


**Limitations:**

The authors are clear about the weakness of the model in capturing aspects of the data. But they should discuss possible alternative approaches. There should be some discussion of the limitations of this paradigm and what it means for investigating structure learning in general.

---

> ### Author Rebuttal · Authors · 2023-08-09
>
> We thank the reviewer for the positive feedback and constructive suggestions.
>
> ***Reply to "Implication of the results to more naturalistic scenarios"***
>
> Our experimental design is an abstraction of many cognitive tasks that require density estimation, or finding statistical patterns from samples. A couple of motivating examples were given in lines 22-24. In addition, human’s ability to learn a probability distribution from samples is demonstrated in many prior works (Ernst & Bank 2002; Hills et al. 2002; Trommershauser et al. 2003; Kording & Wolpert 2004), but how they acquired such distributions are known.
>
> From a broader perspective, the current paper may help address why humans construct a biased perception of the uncertain world. For example, Ravignani et al (2016) [\*] found that, after several rounds of laboratory music evolutions, participants’ reported distribution of rhythms became much more clustered than the original uniform distribution (Figure 2 in their paper). In other words, they found that humans exhibit an over-clustering tendency in the cultural evolution of music, which is similar to our empirical findings. Notably, Ravignani et al did not provide a cognitive model to explain why or how such over-clustering occurs in music evolution, and our DEF approach offers a promising theoretical method to analyze those results. Moreover, the overestimation of clusters we found may be related to the perception of illusory causal relationships in human superstitious thinking [\*\*]. Future work could consider extending our DEF to the construction of high-dimensional uncertainty representation and the emergence of causal connections among different dimensions.
>
> [\*] Ravignani, A., Delgado, T., & Kirby, S. (2016). Musical evolution in the lab exhibits rhythmic universals. Nature Human Behaviour, 1: 0007. doi:10.1038/s41562-016-0007.
>
> [\*\*] Matute, H., Blanco, F., Yarritu, I., Díaz-Lago, M., Vadillo, M. A., & Barberia, I. (2015). Illusions of causality: How they bias our everyday thinking and how they could be reduced. Frontiers in Psychology, 6, 1–14.
>
> ***Reply to "Discussion of alternative models"***
>
> Please see the general response. Our previous statement about the model fitting performance was too pessimistic. Further results in the response PDF shows that our best model—the economical DEF captures the human data in many aspects other than cluster number and sd,  and approaches the upper limit of test-retest reliability in predicting $K$.
>
> ***Reply to "Parameter recovery"***
>
> We ran the suggested parameter recovery experiments and show the results in the response PDF. Given randomly chosen parameters for the full model, we generate 100 sets of synthetic stimuli, reset the parameters to new random values, and then fit the parameters on the synthetic dataset using the procedure described in the main paper. The results show that the recovered parameters are largely consistent with the random initial values (Panel K in response PDF, the average correlation between the source parameters and the fitted parameters is 0.84). We will include parameter recovery in the Appendix.
>
> ***Reply to "Overparameterization"***
>
> The opposite effects of decay rate and prior variance offer an explanation for the observed reported distribution on K, but these two parameters do not constitute overparameterization of the model for this given dataset. This is because the prior variance is also fit by the reported cluster variance in the dataset and is thus well-specified. There are only weak correlations between these two fitted parameters (Experiment 1: -0.12, Experiment 2: -0.05, Experiment 3: -0.05). Further, the parameter recovery experiments also indicate that these two parameters are identifiable given the data.
>
> ***Reply to "More clusters with sample size despite having a decaying alpha"***
>
> We clarify that the effect of a decaying alpha is to decrease the *tendency* toward creating a new cluster, rather than decreased the cluster count itself. As such, even though $\alpha_t$ becomes small at large sample size $t$, the model size $K_t$ can still increase.
>
> ***Reply to "Additional behavioral experiments to test exchangeability"***
>
> The suggested experiment that tests for exchangeability by well-designed stimulus order is an interesting direction that we are currently exploring. The exact quantifiable effects of order-dependence are not directly obvious and may be intertwined with other cognitive processes, such as primacy, recency, and hindsight effects. Various hypotheses can lead to different predictions that are best tested under carefully manipulated stimulus order, opening up a large design space for exploration. We thus defer this to future work.

---

> > ### Comment · Reviewer_Ds9y · 2023-08-15
> >
> > Thanks for the clarification. I am now slightly more confident about my original rating so I increased my confidence score.

---

### Author Rebuttal · Authors · 2023-08-09

We thank all reviewers for their careful read of the paper and helpful feedback. We are glad that all five reviewers gave fair and comprehensive summaries, indicating that the paper is mostly well written, as also stated by four Rs (Ds9y, Xmkb, ti72 & 8bgA).  **All** Rs acknowledged that this work is interesting or novel. Three Rs (**Ds9y**, **Xmkb** & **ti72**) also praised the modelling work as clean, rigorous, non-trivial, and valuable.

Meanwhile, most of the concerns by the Rs can be resolvable with more modelling findings (2 alternative priors for batch ICP), and richer comparisons between human behaviors and model predictions (11 figures in PDF), which we provide in this rebuttal and will incorporate in a revision. We believe these will resolve most of the weaknesses and questions of the Rs and improve the quality of the manuscript.

***How well does the model produce human data? (Ds9y, Xmkb, ti72)***

We initially wrote “the prediction errors on the number of clusters and cluster variance are still large” in the last paragraph as one of the limitations. We now found this to be overly pessimistic. More comprehensive comparisons between the human data and model predictions (see response PDF) indicate:
1. Our economical DEF model produces human patterns in many aspects, not only in the measures the loss function was optimized for (i.e., the number, mean, sd, and weight of clusters), but also in the moments of the whole distribution (mean, sd, and skewness) and the co-variance of different measures (e.g., how the reported cluster sd decreases with cluster number and eccentricity, but increases with cluster weight).
2. The economical DEF performs roughly as well as the participants themselves in predicting the number of clusters $K$. In experiment 2, there are repeated trials with identical stimuli ($\mathbf{x}_T$). Using these trials, we show in Panel C of the PDF that in repeated trials participants reported a different number of clusters ($K$) just below 50% of the time. Similarly, our model could predict $K$ with accuracy around 0.5, very close to the participants themselves. The predictive power of the best model is thus limited by the stochasticity of human behavior.

***Can the data be explained by simpler models? (Ds9y, Xmkb, 8bgA)***

The complexity of our economical DEF as a model of our participants is not as high as it appears to be. What we assume for the participant is that they only use a *single* latent particle *at each time step* to assign the new sample to a cluster. The large number of Monte Carlo simulations is only needed for us as experimenters to fit the model, to marginalise out the latent cluster assignments in the participant’s mind and unobservable to us.

Conceptually, the CRP-GMM ICP is consistent with a simple heuristic. First, whether a new cluster should be introduced depends on a) the prior tendency to add a new cluster, described in Eqn 1; and b) how well the incoming sample is captured by the current distribution, measured by the Student’s t distribution in lines 177-178. Second, the Gaussian distribution only requires the mean (location) and variance (width), which the participants specify for each cluster.

To further validate whether the key computations in the CRP-GMM ICP are indeed necessary, we compared 11 simpler models in the paper (Appendix B.4.1). Below are examples that implement a static or less adaptive prior:
1. The batch baseline replaces the sequential prior with a Poisson distribution and changes the structure of the ICP to a static prior;
2. The No Counting Prior ablation replaces the counts of the clusters (core feature of CRP) with the average count;
3. No Distribution Prior ablation removes the conjugate prior over the Gaussian distribution when evaluating the likelihood of the new sample;

All these simpler models (see Appendix B.4.1 for the full list) led to worse fits than CRP-GMM ICP models (the exchangeable DEF and the economical DEF).

***Intuition and motivation for the aleatoric component (8bgA, ziH7)***

The core motivation for having the aleatoric component is to provide a well-defined likelihood function that is also cognitively plausible. Though the rational component induced by the ICP is stochastic and can in theory generate almost all behaviors with nonzero probability, we found empirically that ICP alone cannot practically predict the correct K in all trials for all subjects (i.e., likelihood is near zero). The issue lies in (as 8bgA commented) that subjects may not report exactly the clusters from their inference, due to memory and motor noises. The aleatoric component provides a highly-structured noise model to accommodate such noises (Eqns 5 & 6), which allows our model to better capture human data.

In its technical challenge and solution, the aleatoric component resembles the classic example of drift-diffusion models (DDMs): the behavioral quantities are multi-dimensional, mixed with discrete and continuous variables; to jointly model these quantities, one can first model the discrete distributions, and then construct a conditional distribution for the continuous variables given the discrete. For DDMs, the continuous response time distribution is often conditioned on a discrete correct/wrong label. Similarly, our DEF places a conditional distribution over the discrete $K$. The rest part of the DEF is more complicated than DDMs, because
1. the support of the predicted continuous variables (w, m, sd) depends strongly on the number of predicted clusters, and
2. the permutation invariance of the parameters (w, m, sd) in defining the density function.

---

> ### Author Response · Authors · 2023-08-10
> **The correct version of figure legends for response PDF**
>
> Dear Reviewers,
>
> We just found that the latest version of response PDF we submitted to OpenReview (at 12:43 am EDT) was mistakenly overwritten by an older version of the file. The new PDF file (we had uploaded) differs from the old file (what you see in the system) mainly in the figure legends, which we have revised to improve readability. The following is the new figure legends, for your reference. Sorry for the inconvenience!
>
> **A** & **B**, the enhanced plots for Figure 1B & D in the main text. The grey dots are newly added. Each dot is for one subject in one sample-size condition. **C**, the proportion that the reported numbers of clusters are different when two trials have (1) different distributions, (2) same distribution but different samples, or (3) same distribution and samples. **D**–**F**, the reported moment versus the sample moment, with data (left sub-panels) contrasted with model prediction (right sub-panels). Three rows are for mean, sd, and skewness. In each panel, the grey heatmap denotes the distribution of responses of individual trials (collapsed across subjects). The 5 dots and error bars denote average response and standard error across subjects in 5 local data bins. The line denotes the regression line, with shading representing 95% confidence interval. **D** & **E** are respectively for the sample-size conditions of 10 and 70 in Experiment 1. **F** is for Experiment 2. **G**–**J**, the detailed patterns of the reported cluster properties in Experiment 2. **G**, the cluster sd decreased when the number of reported clusters increased. **H**, the cluster sd decreased when the cluster eccentricity (measured by the distance between the cluster center and the global center) increased. **I**, the cluster sd increased when the cluster weight increased. **J**, the relative frequency of reported cluster number in different conditions in Experiment 2. **K**, the quality of model recovery, measured by the correlations between the source parameters and the fitted parameters. **To illustrate that our model can capture all behavior patterns mentioned above, we use red color to highlight the behavior of the fitted model agent in panel D-J.**

---

### Decision · Program_Chairs · 2023-09-21

**Decision:**

Accept (poster)

**Comment:**

This paper studies how humans estimate probability distributions and presents several studies where human subject observe iid samples and are asked to estimate a mixture model distribution. The paper hypothesizes that the inconsistency between the human models and the ground truth is due to limited memory. The paper then models this process using a nonparametric Bayesian model that combines a rational component with an aleatoric component, and then evaluates this model on the data tasks.

Reviewers found the paper to contain novel contributions including a behavioral task and a new computational model. During the discussion phase, reviewers appreciated the authors rebuttal and found it clarified many questions. We encourage the authors to revise the manuscript in order to clarify many of the questions that arose during the discussion phase, including the three main questions discussed in the rebuttal below.